# The juxtamembrane linker of synaptotagmin 1 regulates Ca²⁺ binding via liquid-liquid phase separation

Nikunj Mehta[1], Sayantan Mondal [2], Emma T. Watson [1], Qiang Cui [2] &
Edwin R. Chapman [1] ✉

Synaptotagmin (syt) 1, a Ca²⁺ sensor for synaptic vesicle exocytosis, functions in vivo as a multimer. Syt1 senses Ca²⁺ via tandem C2-domains that are connected to a single transmembrane domain via a juxtamembrane linker. Here, we show that this linker segment harbors a lysine-rich, intrinsically disordered region that is necessary and sufficient to mediate liquid-liquid phase separation (LLPS). Interestingly, condensate formation negatively regulates the Ca²⁺-sensitivity of syt1. Moreover, Ca²⁺ and anionic phospholipids facilitate the observed phase separation, and increases in [Ca²⁺]ᵢ promote the fusion of syt1 droplets in living cells. Together, these observations suggest a condensate-mediated feedback loop that serves to fine-tune the ability of syt1 to trigger release, via alterations in Ca²⁺ binding activity and potentially through the impact of LLPS on membrane curvature during fusion reactions. In summary, the juxtamembrane linker of syt1 emerges as a regulator of syt1 function by driving self-association via LLPS.

It is well established that Ca²⁺ binding to the synaptic vesicle (SV) protein, p65[1]/synaptotagmin (syt) 1[2] triggers rapid, synchronous synaptic vesicle exocytosis[3–5]. Given its importance in synaptic transmission, syt1 has been studied in detail. Structurally, it consists of an N-terminal luminal domain, a single transmembrane helix, and a juxtamembrane linker followed by tandem C2-domains (designated C2AB) that are connected by a short, flexible segment (Fig. 1a). The tandem C2-domains are thought to bind -five Ca²⁺ ions via the acidic side chains of five conserved aspartate residues in the Ca²⁺-binding loops in each C2-domain[6–8]. Ca²⁺ binding is greatly facilitated by anionic phospholipids, namely phosphatidylserine (PS) and phosphatidylinositol 4,5-bisphosphate[8–10]. In response to Ca²⁺, the C2-domains of syt1 penetrate membranes[11–13] to trigger fusion[14–16]. Syt1 has also been shown to regulate folding transitions in SNARE proteins[17,18], but the relevance of these interactions remains more controversial[19,20]. In addition, syt1 clamps spontaneous release[14,18,21] and plays a key role in the docking[22–24], priming[25] and endocytosis[26–28] of SVs. Here, we focus on the poorly characterized juxtamembrane segment of this protein.

Since syt1 was cloned -three decades ago[2], it has been notoriously difficult to express and purify its entire cytoplasmic domain (residues 80–421) for in vitro studies. As a result, since the early 1990s, most biochemical studies made use of a truncated form of the protein, lacking the N-terminal region of the cytoplasmic domain (residues 80–95)[29]. The truncated protein (residues 96–421) was far more soluble and had much higher yields than the complete cytoplasmic domain. However, there was an indication that the complete juxtamembrane linker, starting at residue 80, might enhance the binding of Ca²⁺ to the C2A domain, in a C-terminally truncated construct lacking the C2B domain[30]. These findings suggested that the linker serves more than a tethering function.

Furthermore, the juxtamembrane linker has been implicated in the oligomerization of syt1[31]. This is of particular interest, given that early studies using *Drosophila* revealed intragenic complementation between several distinct mutant *syt1* alleles, strongly suggesting that syt1 functions in vivo as an oligomer[21]. More recently, electron microscopy (EM) studies from the Rothman group revealed that syt1 forms ring-like oligomers on lipid monolayers, and these structures

[1]Howard Hughes Medical Institute, Department of Neuroscience, University of Wisconsin–Madison, Madison, WI 53705, USA. [2]Department of Chemistry, Boston University, Boston, MA 02215, USA. ✉e-mail: chapman@wisc.edu

were disrupted by $Ca^{2+}$ [32,33]. In contrast, other EM studies have shown that syt1 forms heptameric barrels in the presence of $Ca^{2+}$ [34]. In even greater contrast, electron paramagnetic resonance and nuclear magnetic resonance studies suggested that recombinant syt1 does not multimerize at all [35,36]. Finally, density gradient experiments indicated that syt1 is a monomer that dimerizes in response to $Ca^{2+}$ [37], or is a constitutive dimer [38] or a tetramer [8]. Binding assays demonstrated direct interactions between copies of an amino-terminal fragment of syt1, but in these experiments the order of oligomers was unknown [31]. Owing to different experimental conditions and techniques, it remains somewhat unclear whether and how syt1 oligomerizes.

As a result of these uncertainties, we recently developed the means to purify the complete cytoplasmic domain of syt1 (residues 80–421), and addressed its ability to oligomerize under near-native conditions using atomic force microscopy [39]. We observed self-association, on lipid bilayer surfaces, into patches and large irregular ring-like structures, mediated by the lysine-rich motif within the juxtamembrane linker. Importantly, mutations that disrupted these interactions impaired the ability of syt1 to both trigger robust, synchronous release in neurons and to clamp spontaneous release [39]. Despite this progress, it is still unclear precisely how syt1 self-associates, as the oligomers lack any apparent ordered structure when visualized under native, aqueous conditions.

Here, we delve more deeply into the properties of the juxtamembrane linker of syt1 and address its role in (a) $Ca^{2+}$ sensing activity, and (b) self-association/multimerization. To our surprise, the lysine-rich motif in the juxtamembrane linker acts as a negative regulator of $Ca^{2+}$ binding to the C2-domains of syt1. To explore the mechanism underlying this negative regulation, we made the discovery that the complete cytoplasmic domain of syt1 self-associates via liquid-liquid phase separation (LLPS), and these interactions are mediated by the juxtamembrane segment. LLPS occurs when a homogeneous liquid

phase partitions into two liquid phases (dilute and concentrated phases). It is usually observed in intrinsically disordered proteins (IDPs) that form membrane-less biomolecular condensates [40–42]. Depending on the IDP sequence, LLPS can occur either by the creation of energetically favorable multivalent protein-protein interactions (hydrogen bonding, hydrophobic and electrostatic interaction, π-π stacking, cation-π, etc.) or by the release of unfavorable, preordered, hydration-shell water molecules [43,44]. LLPS has been described for numerous cytosolic proteins and a more limited number of integral membrane proteins [45]. In this work, we characterize the LLPS of syt1 in detail and describe a model in which the juxtamembrane region, and LLPS, might serve to fine-tune the function of syt1 in nerve terminals.

## Results

### Juxtamembrane linker of syt1 inhibits $Ca^{2+}$ binding to the C2-domains

As outlined above, our understanding of the biochemical properties of the complete, intact cytoplasmic domain of syt1 is limited due to challenges with the expression, purification, and solubility of this protein fragment. We successfully addressed these issues by using a SUMO tag to increase solubility; this tag was removed for all of the biochemical experiments described below. In the first series of experiments, we addressed the ability of the complete cytoplasmic domain to bind $Ca^{2+}$. The constructs used for these studies are illustrated in Fig. 1b; fragments starting at residues 80- and 96- are discussed above in the Introduction. We also analyzed a mutant version of the linker, referred to as 80 JuxtaK-142 (See Supplementary Fig. 1a, b for the WT and mutated protein sequence), in which all the lysine residues in the juxtamembrane linker, except one, were substituted with polar residues; these mutations affect the ability of syt1 to self-associate [39]. $Ca^{2+}$ binding was monitored via isothermal titration calorimetry (ITC), as described [46] and shown in Fig. 1c–f. In these traces,

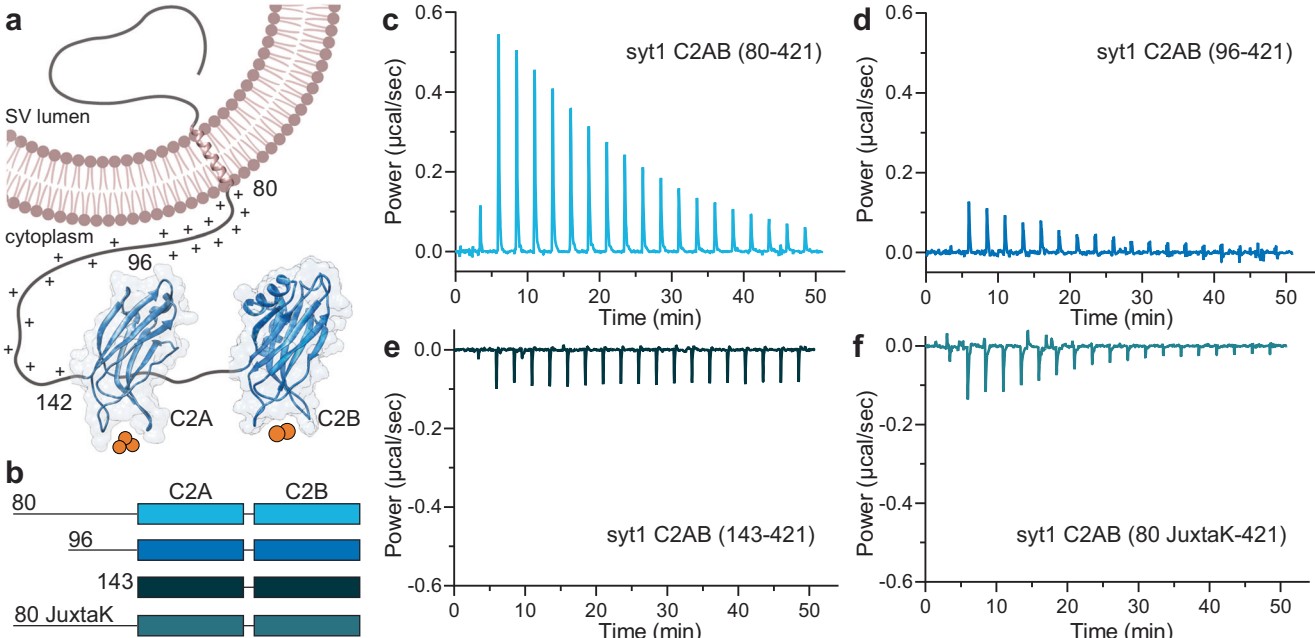

**Fig. 1 | The juxtamembrane linker reduces the $Ca^{2+}$ affinity of the C2-domains of syt1. a** Depiction of full-length syt1 embedded in a synaptic vesicle membrane. The juxtamembrane segment, residues 80–142, contains nineteen lysine residues (indicated by the +), concentrated within residues 80–95. The $Ca^{2+}$-binding C2-domains, C2A and C2B, were rendered using UCSF Chimera and PDB files 1RSY and 1K5W; $Ca^{2+}$ ions are shown as orange spheres. The synaptic vesicle membrane was created using BioRender. **b** Schematic diagram of the syt1 C2AB constructs used for isothermal titration calorimetry (ITC), which includes the entire juxtamembrane

linker (80–142), a truncated linker starting at position 96 (extensively used in the literature), complete removal of the linker, and a mutated JuxtaK linker (80 JuxtaK) in which the lysine residues have been substituted to other polar residues [39]. **c–f** Representative ITC traces showing the heat of $Ca^{2+}$ binding to each of the constructs shown in **b**; $n = 3$. The linker reduces the affinity of C2-domains for $Ca^{2+}$. This effect is largely abrogated in the JuxtaK mutant; moreover, $Ca^{2+}$ binding became exothermic for this mutant linker. Dissociation constants and thermodynamic values are reported in Table 1 and Supplementary Table 1, respectively.

**Table 1 | Dissociation constants (K$_D$ values) for Ca$^{2+}$ binding to the indicated syt1 cytoplasmic domain constructs, measured using ITC**

| Dissociation constant | syt1 C2AB (80–421) | syt1 C2AB (96–421) | syt1 C2AB (80 JuxtaK-421) |
|---|---|---|---|
| # binding sites | 5 | 5 | 4 |
| K$_{D1}$ (μM) | 80.0 ± 3.9 | 23.5 ± 2.1 | 14.1 ± 2.5 |
| K$_{D2}$ (μM) | 190 ± 25 | 210 ± 89 | 24.2 ± 4.4 |
| K$_{D3}$ (μM) | 309 ± 53 | 250 ± 110 | 156 ± 19 |
| K$_{D4}$ (μM) | 1310 ± 125 | 451 ± 100 | 925 ± 150 |
| K$_{D5}$ (μM) | 2900 ± 321 | 2510 ± 430 | |

Data are mean values ± SEM; $n$ = 3.

each injection of Ca$^{2+}$ results in a peak, as heat is either liberated or absorbed. Ca$^{2+}$ injections progress from left to right, with the peak amplitude diminishing as the protein becomes saturated. The data were well-fitted using a five-binding site model for both the complete, intact cytoplasmic domain (residues 80–421) and the widely used truncated 96–421 construct.

Intriguingly, inclusion of the complete linker resulted in a significant reduction in the intrinsic affinity for Ca$^{2+}$ of all five sites compared to the construct lacking residues 80–95. In the case of the highest affinity site, the K$_D$ increased from 23.5 ± 2.1 to 80.0 ± 3.9 μM for Ca$^{2+}$ upon removal of residues 80–95 (Fig. 1c, d, Table 1). Removal of the entire linker, resulting in the 143–421 construct, yielded relatively flat ITC traces due to the cancellation of the endo- and exothermic signals from each C2-domain, respectively, as documented previously[46], precluding a determination of Ca$^{2+}$ affinity (Fig. 1e). We note that the juxtamembrane linker has been studied before, in the context of Ca$^{2+}$ binding to the C2A domain[30]. In contrast to our findings, the authors of the earlier study concluded that the linker enhances the affinity of C2A for Ca$^{2+}$. The basis of this discrepancy is unclear but might involve differences in protein purification. Namely, we employed fast protein liquid chromatography, high salt washes, as well as DNase and RNase treatment, to remove protein and nucleic acid contaminants[34,36]. Neutralization of the lysine residues in the linker segment resulted in exothermic binding, and the data were best-fitted with a four-site model. This mutant yielded the highest measured affinity for Ca$^{2+}$ among all the constructs tested; the K$_D$ of the highest affinity site was 14.1 ± 2.5 μM (Fig. 1f, Table 1). All of the thermodynamic parameters from the ITC experiments are provided in Supplementary Table 1. These findings reveal that the juxtamembrane linker can negatively affect the Ca$^{2+}$ binding activity of syt1 via a process that is disrupted by the lysine mutations. In the next series of experiments, we explore how this might occur.

## Molecular dynamics simulations predict syt1 undergoes LLPS

Analysis of the linker, using the VL-XT algorithm and PONDR software[47], revealed that residues 82-133 form an intrinsically disordered region (IDR; Fig. 2a). These structural predictions were further confirmed using AlphaFold[48,49] (Supplementary Fig. 2). Since IDRs are often involved in LLPS[40–42], we conducted coarse-grained molecular dynamics (MD) simulations and found that the isolated syt1 linker segment stably self-associated, suggesting it may form LLPS droplets (Fig. 2b, Supplementary Movie 1, Supplementary Data 1, 2). Importantly, neutralization of the lysine residues disrupted interactions between linker molecules, and are, therefore, predicted to result in the dissolution of the droplets (Fig. 2c, Supplementary Movie 2, Supplementary Data 3, 4). This effect is quantified in Fig. 2d, where the lysine mutations are predicted to result in the net loss of contacts between different copies of the linker. Next, we performed coarse-grained MD simulations of the cytoplasmic domain of syt1 (Supplementary Movie 3), with and without the juxtamembrane linker, to further

address the role of this segment in LLPS. As shown in Supplementary Fig. 3a–c, syt1 molecules, with an intact juxtamembrane linker, have a higher probability of forming large clusters, while syt1 fragments lacking the juxtamembrane linker formed a larger number of smaller clusters. See Supplementary Data 7–12 for force-field parameter files and GROMACS input files.

We then carried out atomistic MD simulations of the condensate formed by the isolated linker segment and found that the lysine sidechains (residues 80–95) were involved in the formation of multiple long-lived hydrogen bonds (HBs) with glutamate and aspartate residues from different chains (Supplementary Fig. 4a–c). These bridging HBs provide additional enthalpic stabilization along with the other backbone-backbone and backbone-sidechain HBs. To further address the role of charged residues of syt1 in LLPS, we determined the fraction of positive and negative charges, fraction of charged residues (FCR), net charge per residue (NCPR), and sequence charge decoration (SCD) values for the juxtamembrane linker (residues 80–142), as shown in Supplementary Table 2. We obtained an FCR value (0.555) greater than 0.3, and a high negative value of the SCD parameter (−21.5); both are predictive of a 'strong' polyampholyte that is likely to undergo phase separation. These values also suggest the condensates are resilient to increasing salt concentrations[50,51]. These findings further imply a crucial role of the lysine residues in the juxtamembrane linker of syt1 to facilitate LLPS.

## The juxtamembrane linker of syt1 mediates LLPS in vitro

We then went on to empirically examine potential droplet formation. The constructs in Fig. 3a are syt1C2AB-GFP fusion proteins that include a complete, truncated, or mutated juxtamembrane linker, whereas the constructs in Fig. 3b are the isolated WT or mutated (JuxtaK) juxtamembrane linkers, again fused to GFP. We emphasize that we used a monomeric, superfolder version of GFP (see Methods) that does not promote aggregation when fused to other proteins[52,53]. This version of GFP has been used to study LLPS in numerous studies[54–57], and was used, in an untagged form, as a negative control in our studies. The formation of droplets was assessed via fluorescence microscopy, in the presence of 3% PEG 8000, a crowding agent commonly used to study LLPS. Notably, the complete cytoplasmic domain of WT syt1 (C2AB (80–421)-GFP) formed droplets. In stark contrast, no droplet formation was observed using the truncated or mutated juxtamembrane linker constructs (Fig. 3c). These findings reveal that the lysine residues in the juxtamembrane linker are crucial for the cytoplasmic domain of syt1 to undergo LLPS. To confirm these results, we labeled both the complete and truncated cytoplasmic domain of syt1 (residues 80–421 and 143–421, respectively) with an organic dye (fluorescein) at native Cys residues 82 and 277. As expected, we observed droplet formation with the complete cytoplasmic domain of syt1 but not for the truncated protein lacking the linker (Supplementary Fig. 5). To assess whether the juxtamembrane linker alone is capable of undergoing LLPS, we subjected the constructs in Fig. 3b to the same conditions that were used in Fig. 3c. Interestingly, we found that the GFP-tagged juxtamembrane linker (80–142) formed droplets, and this activity was, again, abolished in the JuxtaK mutant (Fig. 3d). Hence, the juxtamembrane linker alone is sufficient to undergo LLPS, in a lysine residue-dependent manner. GFP alone failed to form droplets under any conditions tested (Fig. 3d).

As an independent measure of droplet formation, we conducted dynamic light scattering (DLS) experiments, again using the constructs shown in Fig. 3a, b, and found that WT syt1 C2AB, bearing the juxtamembrane linker, as well as the isolated juxtamembrane linker (residues 80–142), yielded two distinct peaks with diameters corresponding to monomers as well as higher ordered structures. In sharp contrast, a single peak, corresponding to monomers, was observed for the other constructs, as shown and quantified in Supplementary Fig. 6a, b, and Supplementary Table 3, respectively. These

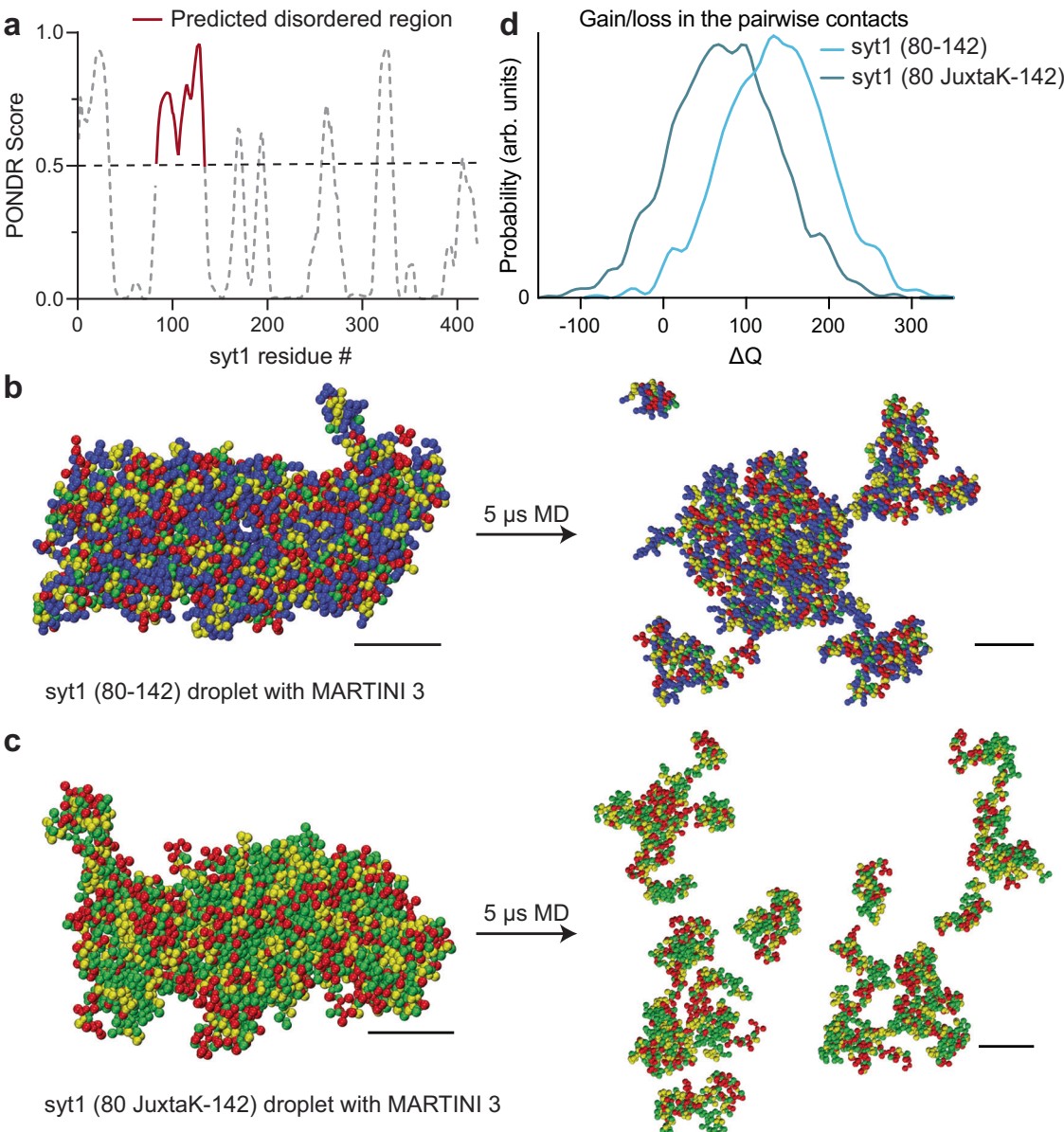

**Fig. 2 | MD simulations predict that the intrinsically disordered region (IDR) in the juxtamembrane linker of syt1 mediates liquid-liquid phase separation (LLPS). a** Analysis using the VL-XT algorithm and PONDR software reveals that the segment comprising of residues 83–133 in the juxtamembrane linker of syt1 has a high probability (>0.5) of being disordered. **b** Left: MD simulation of droplet formation and stabilization by the isolated syt1 (80–142) juxtamembrane linker. Right: The droplet remains intact after 5 μs of simulation. All molecules stay in the dense phase with little branching in the system. **c** Left: Same as **b** Left but for the isolated syt1 (80 JuxtaK-142) mutant linker. Right: The droplet does not retain its shape and dissolves within 5 μs of the simulation protocol. Scale bars, 4 nm. (See Supplementary Movies 1, 2 for the movies). **d** Distribution of the gain/loss in the number of contacts (ΔQ) between copies of either the WT or JuxtaK mutant linker. The lysine substitutions shifted the distribution to the left, indicating a loss in the number of pairwise contacts compared to the same number of isolated chains. In panels **b** and **c**, the color code for the amino acid residues is as follows: blue, positive charge; red, negative charge; yellow, non-polar; green, polar. An explicit solvent coarse-grained molecular dynamics simulation with a reparametrized MARTINI v3.0 force field was used in this analysis. The ionic strength of the buffer was 100 mM NaCl. Each MD simulation is a result of $n = 2$ trajectories. Source data are provided as a Source Data file.

findings are consistent with the microscopy experiments in Fig. 3c, d. Since our DLS approach cannot discriminate between sizes above 1 μm, we returned to microscopy to establish the relationship between [syt1 C2AB (80–421)-GFP] and droplet size. As expected, based on LLPS of various proteins[58], we observed that droplet size increased with protein concentration (Supplementary Fig. 7a–e). Because the droplet number is confounded by droplet-droplet fusion, this parameter was not further analyzed.

Next, to further study the effect of molecular crowding, we characterized droplet formation by the constructs used in Fig. 3a, b, as a function of increasing w/v% of PEG 8000. We again observed that

syt1 C2AB (80–421)-GFP readily formed droplets, even at low protein (0.3 μM) and PEG 8000 (1%) concentrations (Fig. 3e). The isolated linker, GFP-syt1(80–142), also formed droplets, albeit at higher protein and PEG 8000 concentrations (Fig. 3f). Furthermore, we analyzed the syt1 C2AB constructs with truncated or mutated juxtamembrane linkers, as well as isolated JuxtaK mutant linker, and found that none of these constructs formed droplets, even at high protein and PEG 8000 concentrations (Supplementary Fig. 8a–c). These additional findings, combined with results in Fig. 2, provide compelling evidence that the lysine-rich motif in the juxtamembrane linker of syt1 predominantly mediates LLPS.

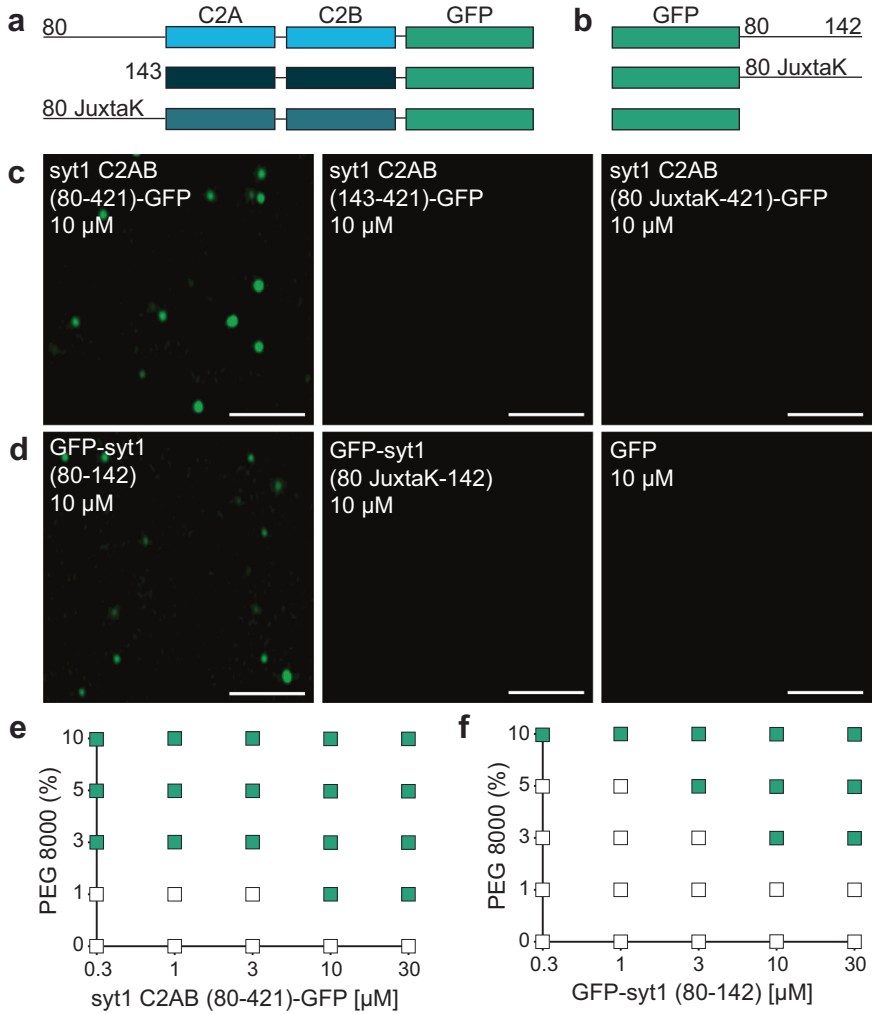

**Fig. 3 | The IDR region of syt1 is necessary and sufficient for LLPS.** Schematics of the syt1 C2AB-GFP **a** and isolated GFP-tagged juxtamembrane linker **b** fusion proteins used to study LLPS. **c** syt1 C2AB (80–421)-GFP and **d** GFP-syt1 (80–142) form droplets in a buffer of physiological salt and 3% PEG 8000, whereas syt1 C2AB lacking this linker, or in which the linker has been mutated (JuxtaK), fails to form droplets; $n = 3$. **e, f** Phase diagram of syt1 C2AB (80–421)-GFP and GFP-syt1 (80–142) with varying protein and PEG 8000 concentrations. The buffer condition was 25 mM Tris-HCl (pH 7.4), 100 mM NaCl. Green squares indicate the appearance of droplets, whereas white squares indicate no droplet formation; $n = 3$. (Scale bars, 3 μm).

### Fusion and FRAP of syt1 droplets

To further explore whether syt1 C2AB (80-421)-GFP droplets we observed correspond to bona fide LLPS, we examined whether they fuse with each other. Figure 4a shows a time series documenting the fusion of two protein droplets over a span of 70 s (see Supplementary Movie 4 for the video). Furthermore, we conducted fluorescence recovery after photobleaching (FRAP) experiment on the syt1 droplets; a time series of representative FRAP images are shown in Fig. 4b (see Supplementary Movie 5 for the video), clearly demonstrating recovery. The fluorescence recovery curve was best fitted with a hyperbolic function, yielding a $t_{1/2}$ of $64 \pm 2$ s (Fig. 4c; see Methods for analysis details), which is characteristic of the protein mobility in bona fide LLPS droplets. Together, these droplet fusions and FRAP results confirm that syt1 C2AB (80–421)-GFP undergoes LLPS.

### Impact of ionic strength, pH, anionic lipids, and Ca²⁺ on syt1 droplets

To gain further insights into the interactions that mediate syt1 LLPS, we explored the roles of ionic strength, pH, soluble 6:0 PS, and Ca²⁺. As the ionic strength was increased from 100 mM to 1 M NaCl, the normalized fluorescence intensity of syt1 C2AB (80–421)-GFP droplets decreased sharply, suggesting that electrostatic interactions play an important

role in droplet formation. In fact, the protein droplets were completely dissolved at 1 M NaCl (Fig. 5a). We also examined this effect via MD simulations of the linker segment using the MARTINI v3.0 force field. Equilibration at 400 mM NaCl partially dissolved the cluster of linkers within the 5 μs simulation (Supplementary Fig. 9a, b, Supplementary Movie 6, Supplementary Data 5, 6). These findings are quantified in Supplementary Fig. 9c, which shows a left-shift in the distribution of loss in the number of contacts between isolated syt1 juxtamembrane linkers at high ionic strength. To further validate these results with the complete syt1 cytoplasmic domain, we, again, performed DLS analysis of syt1 C2AB (80–421)-GFP and observed primarily a single peak corresponding to a monomeric state of the protein at higher ionic strength, as compared to the two peaks observed at lower ionic strength (Supplementary Fig. 9d, Supplementary Table 4). Next, we assessed the effect of pH on the stability of these droplets and found they were the most stable at pH 7 and formed less efficiently at both higher and lower pH values (Fig. 5b). Finally, since syt1 triggers exocytosis by binding Ca²⁺ and anionic phospholipids such as PS[14–16], we explored the effects of [Ca²⁺] and a soluble form of PS, 6:0 PS, on droplet formation and stability. Interestingly, 6:0 PS alone induced the formation of syt1 (80–421)-GFP droplets in a dose-dependent manner, and this was enhanced by the inclusion of 3%

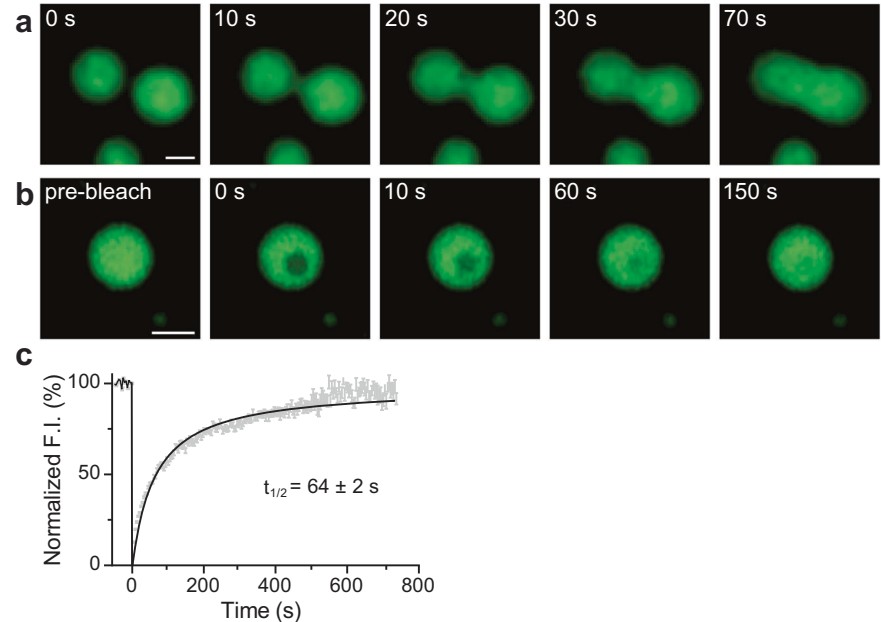

**Fig. 4 | Syt1 droplets fuse with each other, and recover after photobleaching.**
**a** Time series showing two droplets of syt1 C2AB (80–421)-GFP (3 μM protein, 3% PEG 8000) fuse and relax into a larger droplet. **b** Photobleaching a syt1 C2AB (80–421)-GFP droplet and subsequent fluorescence recovery, shown in a representative time series. (See Supplementary Movies 4, 5 for the movies) **c** Quantification of the FRAP experiments from $n = 6$ independent experiments, each

containing 6–10 bleached droplets (total $n = 47$). The data were fitted with a hyperbolic function using Graphpad Prism (solid line); error bars represent SEM. The inset shows the $t_{1/2}$ of the fluorescence recovery. The droplet and bleached region diameters were 2–2.5 μm and 1.6 μm, respectively. Source data are provided as a Source Data file. (Scale bars, 2 μm).

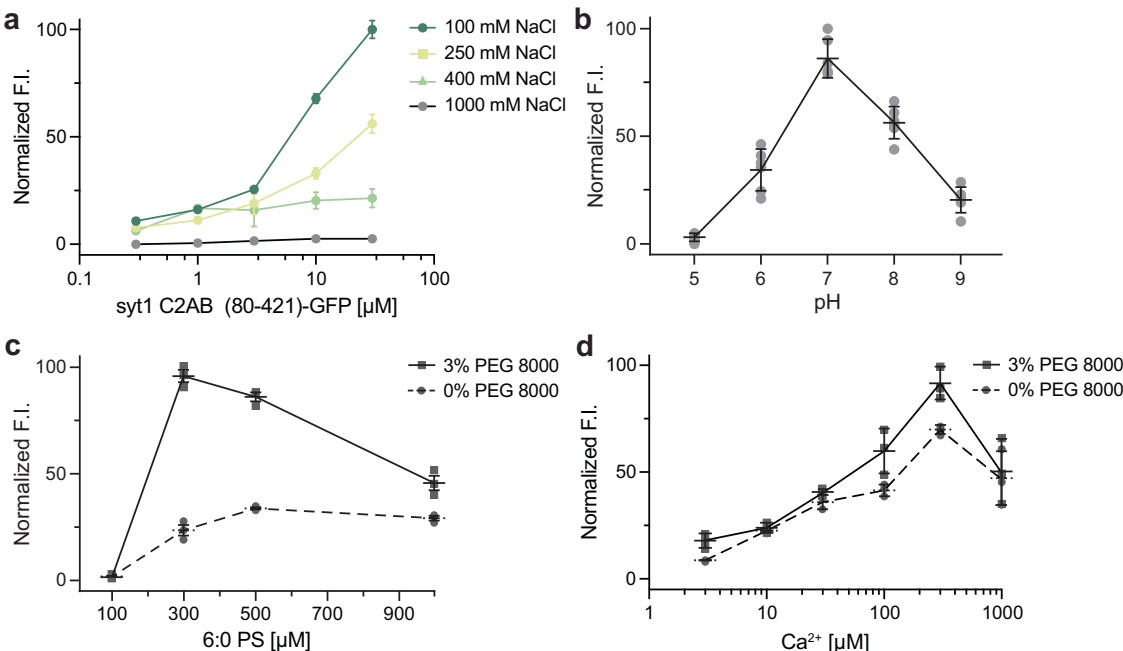

**Fig. 5 | Examination of droplet formation and dissolution of syt1 C2AB (80–142)-GFP as a function of ionic strength, pH, soluble [6:0 PS], and [Ca²⁺].**
**a** Normalized fluorescence intensity (F.I.) of droplets were analyzed as a function of increasing salt and protein concentration (3% PEG 8000). **b** Same as **a**, but as a function of pH; $n = 3$ fields of view and error bars represent SEM. The buffers for

different pH solutions (described in Methods) also contained 100 mM NaCl and 3% PEG 8000. **c** Same as **a**, but as a function of [6:0 PS] in 25 mM Tris-HCl (pH 7.4), 100 mM NaCl, with and without 3% PEG 8000; $n = 3$ fields of view and error bars represent SEM. **d** Same as **c**, but as a function of [Ca²⁺] buffered with EGTA, with and without 3% PEG 8000. Source data are provided as a Source Data file.

PEG 8000 (Fig. 5c). These findings agree with previous work showing that 6:0 PS induced the self-association of syt1 (80–421) by DLS[39]. We observed a similar trend when we titrated the [Ca²⁺] and found that syt1 (80–421)-GFP droplet formation was facilitated in a dose-dependent manner (Fig. 5d).

## Syt1 undergoes LLPS in cells
Next, we sought to determine whether syt1 can form droplets in living cells. Upon overexpression, droplet formation by syt1 C2AB (80–421)-GFP was observed in the HEK293T cells; as shown in Fig. 6a, ~5–10 droplets formed in the cytoplasm of each cell. Consistent with our

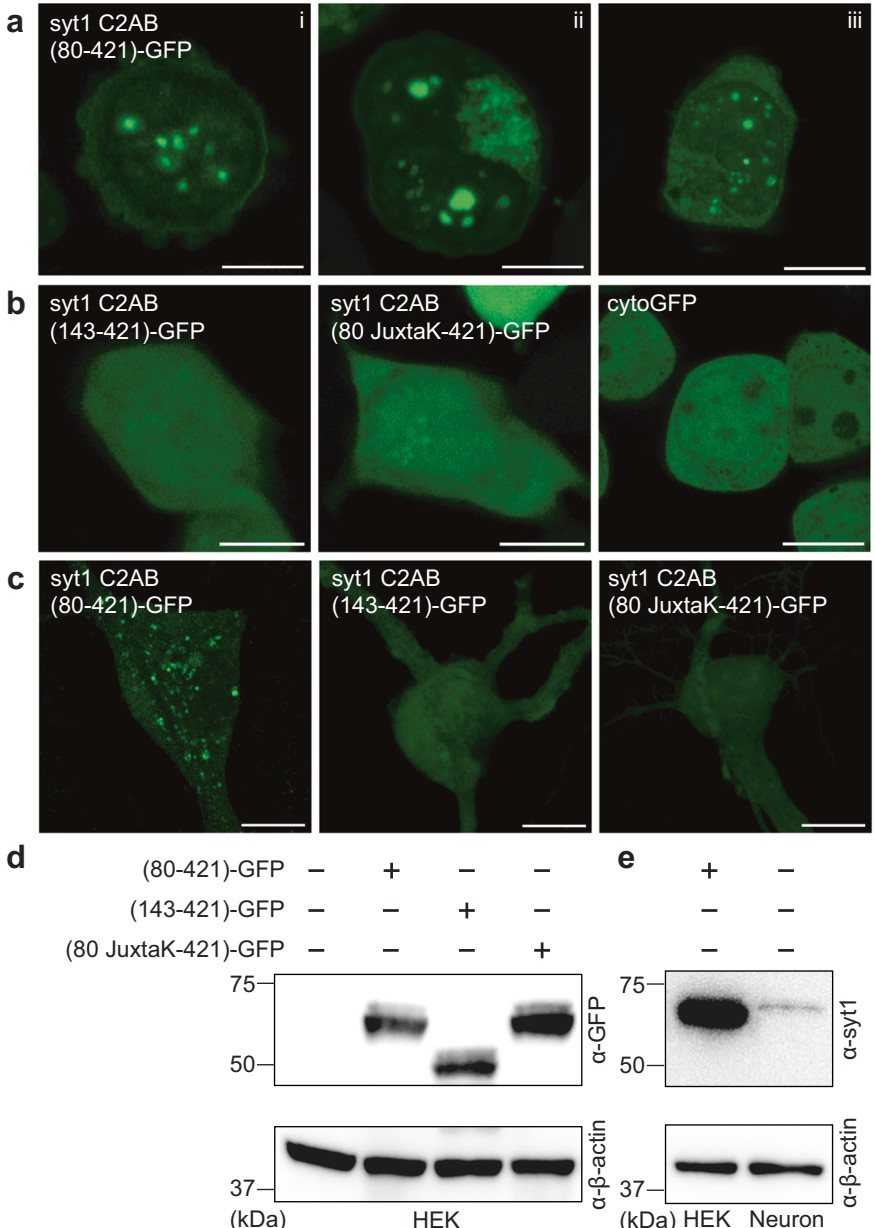

**Fig. 6 | Syt1 forms droplets in HEK293T cells and cultured rat hippocampal neurons. a** Three representative super-resolution fluorescence images of HEK293T cells overexpressing syt1 C2AB (80−421)-GFP showing protein droplet formation 24-48 h post-transfection (Scale bars, 10 μm). **b** Same as **a** but with overexpressed syt1 C2AB (143−421)-GFP, syt1 C2AB (80 JuxtaK-421)-GFP, and cytoGFP. These constructs fail to form protein droplets (Scale bars, 10 μm). **c** Representative super-resolution fluorescence images of rat hippocampal neurons overexpressing transfected syt1 C2AB (80−421)-GFP, syt1 C2AB (143−421)-GFP, and syt1 C2AB (80 JuxtaK-421)-GFP. The first construct forms protein droplets, whereas the other two fail to do so; *n* = 3 (Scale bars, 10 μm). **d** Immunoblot of HEK293T cell lysates with WT and the overexpressed syt1 C2AB constructs described in **a**, stained with an anti-GFP antibody. β-actin served as a loading control. **e** Immunoblot to estimate syt1 C2AB (80−421)-GFP expression levels in HEK293T cells compared to endogenous syt1 in cultured rat hippocampal neuronal lysates, probed using an anti-syt1 antibody. β-actin again served as a loading control; *n* = 2. Uncropped blots and source data are provided as a Source Data file.

in vitro biochemical assays (see Fig. 3c), truncation of the linker, or substitution of the lysine residues within the linker, abolished droplet formation (Fig. 6b). A cytosolic GFP construct (cytoGFP) served as a control and did not form droplets. We then repeated these experiments in hippocampal neurons and observed the same trends (Fig. 6c). For completeness, we addressed the degree of overexpression in these cell-based experiments via immunoblot analysis. HEK293T cells lysates, overexpressing the syt1 C2AB constructs, were probed with an anti-GFP antibody (Fig. 6d). To compare our overexpression levels of syt1 C2AB (80−421)-GFP in HEK293T cells with the levels of endogenous syt1 in cultured rat hippocampal neurons, we subjected equal amounts of lysates to immunoblot analysis using an anti-syt1 antibody

(Fig. 6e). Densitometry revealed that the level of overexpression in HEK293T cells was ~6-fold greater than in cultured rat hippocampal neurons, after correction for the ~80% transfection efficiency in HEK293T cells (we note that the transfection efficiency in cultured rat hippocampal neurons was too low to estimate over-expression levels in this cell type). To further characterize these droplets, we examined their potential dissolution using 1,6-hexanediol, an aliphatic alcohol that interferes with weak hydrophobic interactions within droplets. Indeed, syt1 droplets dissolved (partially or completely) upon treatment with 1,6-hexanediol in both HEK293T cells and neurons, as shown by representative images in Supplementary Fig. 10a, b. Together, these experiments demonstrate that the cytoplasmic domain of

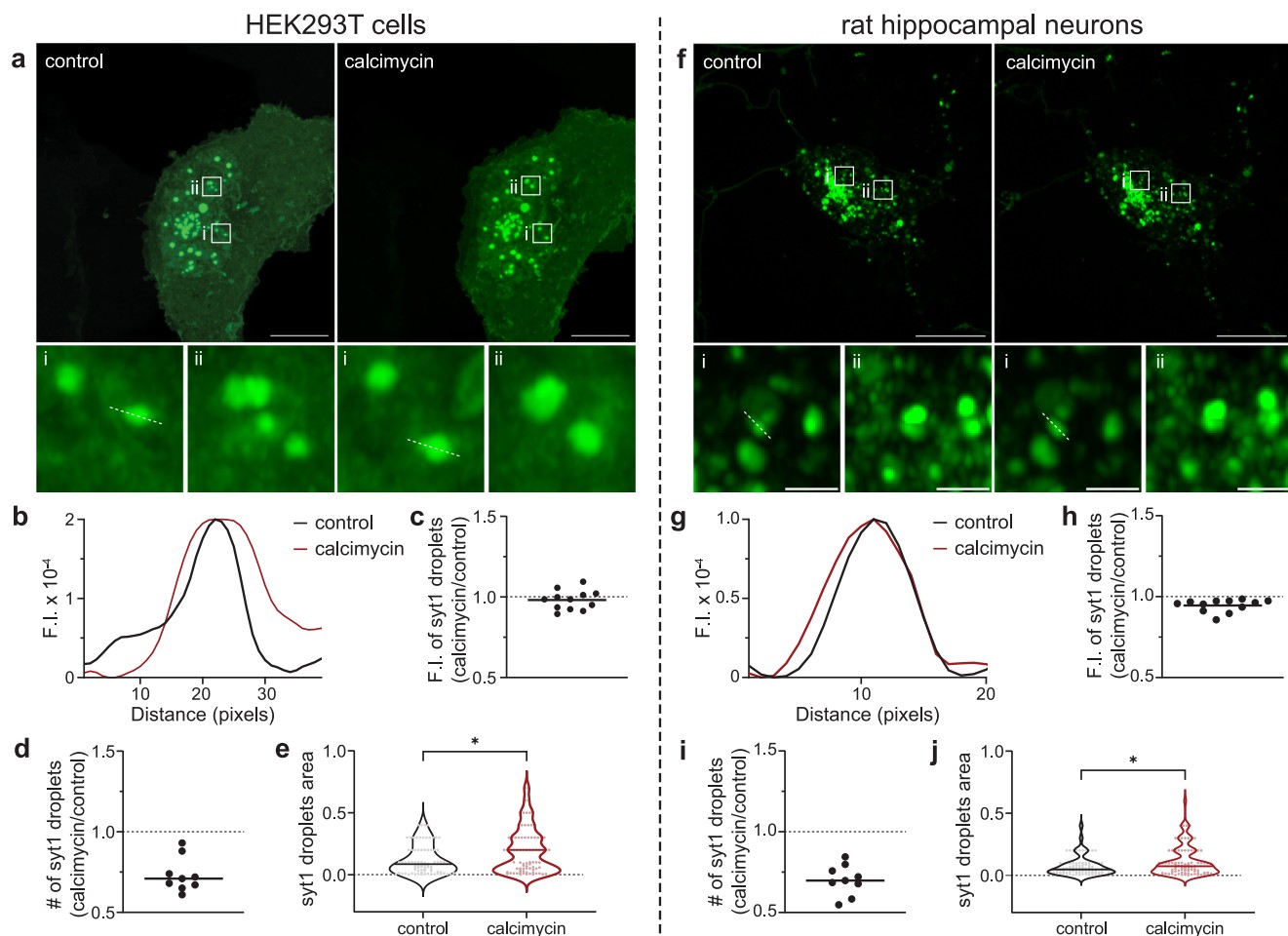

**Fig. 7 | Ca²⁺ influx, via an ionophore, drives syt1 droplet fusion in HEK293T cells and cultured rat hippocampal neurons.** Left: HEK293T cells; Right: cultured rat hippocampal neurons. **a**, **f** Representative images of HEK293T cells/hippocampal neurons overexpressing syt1 C2AB (80–421)-GFP showing protein droplets under control and +calcimycin (a Ca²⁺ ionophore) conditions. Two inset images, shown at higher magnification, reveal the larger droplet size in the +calcimycin condition; $n = 3$. **b**, **g** Representative line scan traces of dashed lines in **a** (i) and **f** (i) measuring fluorescence intensity of syt1 droplets under control and +calcimycin conditions. **c**–**e**, **h**–**j** Plots of syt1 droplet fluorescence intensity (F.I.), number, and area.

Calcimycin treatment had no effect on F.I. but decreased the number of droplets and increased their area (Ratio of number of syt1 droplets under calcimycin vs control conditions: 0.73 ± 0.1 SD (HEK293T), 0.70 ± 0.1 SD (neurons)). A total of nine fields of view were analyzed, and droplet area was measured as described in Methods. Mean values are indicated by the horizontal line in each panel. In panels **e** and **j**, the $p$-values of 0.016 (HEK293T), 0.042 (neurons) were calculated using the Mann–Whitney two-sided test. Source data are provided as a Source Data file. Scale bars, 10 µm. Inset scale bars, 1 µm.

syt1 forms droplets in cells, including neurons where syt1 is normally expressed.

Since Ca²⁺ enhanced LLPS of the intact cytoplasmic domain of syt1 in vitro, we assessed the effect of a Ca²⁺ ionophore, calcimycin, on these droplets in cells. Representative images and line traces show an increase in syt1 (80–421)-GFP droplet area in the presence of calcimycin (5 mM Ca²⁺ in the ECF) as compared to the control condition in HEK293T (Fig. 7a, b) and rat hippocampal neurons (Fig. 7f, g). As described in image analysis (Methods), we carefully assessed the fluorescence intensity, number, and area of syt1 droplets under both conditions for the two cell types. Although the fluorescence intensity of syt1 droplets remained unchanged across the two conditions (Fig. 7c, h), there was a decrease in the number of syt1 droplets in the presence of calcimycin (Fig. 7d, i). Quantification of droplet area revealed a significant increase in area of syt1 droplets under calcimycin conditions in both cell types (Fig. 7e, j), due to fusion between droplets, as detailed below in the Discussion.

We also addressed the question of whether syt1 (80–421)-GFP might interact, via LLPS, with native syt1 on synaptic vesicles, by examining the colocalization of overexpressed syt1 (80–421)-GFP with

endogenous syt1 and synaptophysin (syp, a canonical synaptic vesicle marker) using immunocytochemistry (ICC). Endogenous syt1 was selectively probed using a N-terminal luminal domain antibody. As indicated by the arrowheads in the magnified inset images of Supplementary Fig. 11a, b, syt1 (80–421)-GFP partially colocalizes with endogenous syt1 and syp (Supplementary Fig. 12b, c); the degree of co-localization was limited, as the over-expressed truncated fusion protein was found throughout the neurons. As expected, the two native synaptic vesicle markers were strongly colocalized (Supplementary Figs. 11c and 12a). We note that all neurons express endogenous syp and syt1, but only a handful of cells expressed the transfected syt1 (80–421)-GFP construct, lacking the N-terminal domain. While these co-localization studies are consistent with LLPS-mediated interactions between recombinant and native syt1, future studies will focus on direct measurements of these interactions, and their impact on SV clustering.

In summary, we return to our findings above, that provide insights regarding the interaction of syt1 with Ca²⁺. Namely, the juxtamembrane linker reduces the affinity of the C2-domains of syt1 for Ca²⁺ (Fig. 1c–f), Ca²⁺ enhances syt1 LLPS (Fig. 5d), and increased [Ca²⁺]ᵢ facilitates the

fusion of syt1 droplets in cells (Fig. 7). We therefore examined the ten aspartate residues that coordinate Ca²⁺ ions in the tandem C2-domains of syt1. We assessed the solvent accessible surface area (SASA) values of these residues in a syt1 condensate, as compared to the isolated protein. Six out of the ten aspartate side chains had significantly lower SASA values, resulting in the partial masking of these Ca²⁺-binding residues within droplets (Supplementary Fig. 13). These findings explain how the linker reduces the apparent affinity of syt1 for Ca²⁺; this is an indirect effect mediated by LLPS. How Ca²⁺ facilitates droplet formation and fusion between droplets, remain to be established.

## Discussion

Since the discovery and cloning of syt1[1,2], the Ca²⁺ binding activity of its complete cytoplasmic domain (residues 80–421) had not been examined. This was due to difficulties regarding the solubility of this protein fragment; as a result, most in vitro studies of syt1 made use of a more soluble, truncated fragment of the protein, comprising residues 96–421. Here, using a SUMO-tag during purification to enhance solubility, we isolated the complete cytoplasmic domain of syt1 and conducted ITC experiments to study its affinity for Ca²⁺. The presence of the entire juxtamembrane linker diminished the affinity of the tandem C2-domains for Ca²⁺, compared to a truncated construct comprising residues 96–421 (Fig. 1c, d). Hence, the juxtamembrane linker (residues 80–142) serves as an unexpected negative regulator of Ca²⁺ binding. At the start of the current study, the underlying mechanism remained unclear, but recent work describes how this linker segment, especially residues 80–95, mediates the self-association of syt1[33,39]. Notably, the same truncations or lysine substitutions in the linker that disrupt syt-syt interactions[39] (Supplementary Fig. 6a) also impair the observed negative regulation of Ca²⁺ binding (Fig. 1f). This correlation suggests a model in which the lysine-rich motif regulates Ca²⁺ binding indirectly, by mediating syt1 oligomerization. In this context, it is notable that the lysine residues, which are concentrated between residues 80–95, are partially or highly conserved across species (Supplementary Fig. 1c).

This model hinges on the oligomerization of syt1, yet despite progress on this front, this issue remains murky. As detailed in the Introduction, the propensity of syt1 to oligomerize has been addressed in a myriad of somewhat conflicting studies, and a consensus as to whether and how this protein self-associates has yet to emerge[8,21,31–39]. Here, we provide insights into this question by showing that the juxtamembrane linker, and particularly the lysine-rich motif, mediates self-association by driving liquid-liquid phase separation (LLPS). Namely, coarse-grained MD simulations predicted that the syt1 juxtamembrane linker (residues 80–142) mediates LLPS and that the lysine-rich motif is required for droplet formation. In vitro biochemical studies validated these predictions: the complete cytoplasmic domain of syt1 (residues 80–421) clearly undergoes LLPS. Truncation of, or lysine mutations within the juxtamembrane linker of syt1, abolished the formation of droplets. Moreover, the linker alone is sufficient to undergo LLPS in a lysine-residue-dependent manner, as measured via both microscopy and DLS experiments; thus, DLS may provide a rapid and simple means to monitor LLPS. We further confirmed LLPS via FRAP experiments; the $t_{1/2}$ value for recovery, 64 s, is well within the range of recovery kinetics for other proteins that form droplets[58]. As expected, syt1 droplets were destabilized by high ionic strength and low pH. In contrast, Ca²⁺ and anionic phospholipids, which bind avidly to syt1, facilitate droplet formation. In light of the discovery that syt1 undergoes LLPS, we note atomic force microscopy (AFM) images of the intact cytoplasmic domain of syt1, under aqueous conditions on lipid bilayers, revealed the formation of large ring-like structures and protein patches (Courtney et al.[39]). We propose that these structures reflect syt1 LLPS on the two-dimension surface of membranes.

We also extended our observations to cells and found that over-expression of the complete cytoplasmic domain of syt1 resulted in the formation of droplets in both neurons and fibroblasts. Interestingly,

upon mobilization of $[Ca^{2+}]_i$ with an ionophore, we observed fusion between syt1 condensates, resulting in larger droplets. Droplet growth might also occur due to increased partitioning of protein into the LLPS droplets; at present we are not able discern the relative contributions of these two potential means of droplet growth. Furthermore, the overexpressed complete cytoplasmic domain of syt1 partially colocalized with endogenous syt1 at synapses, potentially via LLPS.

Mutations that disrupt syt1 self-association impair the ability of syt1 to trigger robust, synchronized neurotransmitter release[39], suggesting a positive role for LLPS in exocytosis. Yet, LLPS reduces the Ca²⁺-sensitivity of syt1, and LLPS is enhanced by increasing [Ca²⁺]. Hence, it will be crucial to study the Ca²⁺-dependence for release in neurons that express the JuxtaK mutant form of syt1 that does not undergo LLPS; it is possible that the apparent affinity for Ca²⁺ may be increased, while the activation of syt1 by the bound Ca²⁺ ions is simultaneously impaired. It will also be essential to determine the kinetics of Ca²⁺-promoted LLPS, to see if this occurs on time scales that affect any aspect of the SV cycle. Making this matter even more complex is the observation that droplets can induce either positive or negative membrane curvature, depending on the entropic or enthalpic interactions, respectively, between proteins and lipids[59,60]. Since, again, the JuxtaK mutations impair the ability of syt1 to drive release[39], we favor a model in which syt1 LLPS favors negative curvature to facilitate membrane fusion reactions[61], but this will require further study. The juxtamembrane linker clearly impacts the function of syt1, raising the question of how the activity of this domain is regulated. Indeed, the juxtamembrane linker segment undergoes a number of post-translational modifications, including palmitoylation (C82), acetylation (K98), and phosphorylation (T112, T125, and T128)[62,63]. Whether these covalent modifications modulate the propensity of syt1 to undergo LLPS will be the subject of future studies. All the points discussed in this section are summarized in the schematic diagram shown in Supplementary Fig. 14.

While we have focused on syt1 here, we note that other presynaptic proteins, including α-synuclein, synapsin, and the SV endocytic protein endophilin A1, have also been reported to undergo LLPS[54,64,65]. It will be interesting to determine whether droplets formed by each of these presynaptic proteins interact to potentially form "reaction vessels" that partition SVs in functionally separable pools.

In summary, we propose that the juxtamembrane linker of syt1 plays an important role in regulating the Ca²⁺-sensitivity of its tandem C2-domains, by mediating LLPS via the lysine-rich motif. A vital question to address concerns how LLPS of this integral membrane protein impacts local curvature to alter the energy landscape of fusion reactions. Addressing this question will involve additional MD simulations, in conjunction with reconstitution approaches and cryo-EM to study membrane structure.

## Methods

### Recombinant protein expression and purification

Constructs encoding cytoplasmic fragments of syt1: residues 80–421, 96–421, 143–421, 80 JuxtaK-421 (Fig. 1b), N-terminally fused with his6-SUMO tag, were subcloned into a pET28(a)+ vector and expressed in *E. coli* BL21(DE3) cells. Bacteria were grown to an $OD_{600}$ of 0.6 and induced with 500 μM isopropyl β-D-1-thioga-lactopyranoside (IPTG) (GoldBio, I2481C) for 18 h at 18 °C. Bacteria were collected by centrifugation and sonicated in 50 mM Tris pH 7.4, 1 M NaCl, 5% glycerol plus a protease inhibitor cocktail (PIC) (Roche, 046693132001). One % Triton X-100 (Thermo Fisher Scientific, A16046) was mixed with the lysates at 4 °C for 2 h; lysates were centrifuged at 31,000 × g for 45 min at 4 °C and the supernatant was mixed with nickel-nitrilotriacetic acid (Ni-NTA) beads (Takara, 635653) at 4 °C for 2 h. Beads were washed thrice with wash buffer (lysis buffer with 10 mM imidazole) to remove contaminants[34]; the first wash included DNase and RNase (10 μg/ml each) to prevent nucleic acid-mediated aggregation[34,36]. Finally, beads

were treated with 0.5 μM recombinant SUMO protease (senp2), at 4 °C overnight, to remove the his6 and SUMO tags and liberate the protein of interest. The eluted protein was subjected to an additional purification step using fast protein liquid chromatography (FPLC) and a Superdex 200 Increase 10/300 GL column (Cytiva, 28990944) in Chelex 100-treated (Bio-Rad, 1422832) dialysis buffer (25 mM HEPES pH 7.4, 100 mM KCl). Purified proteins were subjected to SDS-PAGE and protein concentration was determined by running bovine serum albumin (BSA) as a standard.

Constructs encoding GFP-tagged syt1 proteins (as shown in the schematic in Fig. 3a, b) were purified in the same manner, except the proteins were eluted from the beads using 200 mM imidazole in lysis buffer, and samples were dialyzed against Chelex 100-treated dialysis buffer comprising 25 mM Tris pH 7.4, 500 mM NaCl. For organic dye-labeled experiments, syt1 cytoplasmic domain (either residues 80–421 or 143–421) was labeled with fluorescein at Cys82 and Cys277. Twenty-five μM syt1 protein was incubated with a 10-fold excess of fluorescein-5-maleimide overnight at 4 °C. Free dye was removed using a PD-10 desalting column. Note: the GFP tag used throughout this study was msGFP, a monomeric superfolder derivative of GFP[66].

### Isothermal Titration Calorimetry (ITC)

ITC measurements were carried out using a MicroCal iTC200 (Malvern Panalytical, UK). The indicated syt1 proteins (Fig. 1b; 50 μM) were titrated using a 5 mM stock of $Ca^{2+}$. We prepared all solutions in Chelex 100-treated dialysis buffer comprising 25 mM HEPES pH 7.4 and 100 mM KCl; solutions were degassed at 25 °C before each experiment. Titration involved an initial 2 μl addition of $Ca^{2+}$ followed by 18 successive 4 μl additions, with stirring speed at 750 rpm, at 25 °C. We subtracted a blank titration of $Ca^{2+}$ alone to correct for the heat of dilution. Titrations were performed in triplicate and binding constants were determined by curve-fitting to a 4-or 5-site binding model using MicroCal Origin2020 7.0 software.

### Molecular dynamics (MD) simulations

Syt1 intrinsically disordered region (IDR), in the juxtamembrane linker, was modeled using a MARTINI v3.0 coarse-grained (CG) force field[67]. Recently, it was shown that reparametrizing the original force field by increasing the protein-water Lennard-Jones interaction strength ($\varepsilon_{PW}$) can capture the realistic single-chain conformational ensemble of several IDRs[68]. However, the scaling factor ($\lambda$) needs to be determined individually for different IDRs by comparing them to available experimental data. Since the single-chain properties were well-correlated with the peptide/protein phase behavior[43], one can employ the scaling strategy to study LLPS with MARTINI.

The structures of WT syt1 IDR and JuxtaK-mutant IDR were created from their sequences in PyMOL and then converted them into respective CG models and topologies using the *martinize2* code[69]. We randomly inserted fifty single chains into a 30 nm cubic box, followed by the addition of CG water beads (each equivalent to 4 water molecules) and the appropriate number of ions ($Na^+$ and $Cl^-$) to maintain the 100 mM ionic strength. Equilibration of the systems were done for 3 μs with the original MARTINI v3.0 model, where the IDRs phase separated to create droplets. Starting from these, we systematically tuned $\varepsilon_{PW}$ to find a reasonable scaling factor ($\lambda = 1.04$) for which the WT droplet maintained its shape for another 5 μs and the mutant droplet dissolved within 1 μs. Then, the scaled force field was employed to study the fate of the droplet by increasing the ionic strength from 100 mM to 400 mM and found that the droplet dissolved within a few μs. Therefore, the reparametrized MARTINI v3.0 captured two important experimentally observed features: the effect of lysine mutations and salt dependence. In addition to the LLPS simulations, separate isolated single-chain simulations were performed corresponding to each condition. Additionally, MD simulations involving the full cytoplasmic domain (residues 80–421) and the C2AB domain (residues 143–421,

without the linker) of syt1 were performed with the same force field parameters. For each of these simulations, we solvated 20 protein molecules within a 35 nm cubic box.

The equilibration simulations were propagated with a time step of 10 fs and the production simulations with a time step of 20 fs, using the leap-frog algorithm. We used the V-rescale thermostat[70] (with $\tau_T = 1\,ps^{-1}$) at 298 K and the Parrinello-Rahman barostat[71] with isotropic pressure coupling ($\tau_P = 12\,ps^{-1}$) at 1 bar. For initial equilibration purposes, we used the Berendsen barostat[72] with $\tau_P = 6\,ps^{-1}$. The electrostatic interactions were screened with a dielectric constant ($\varepsilon_r$) of 15 within a cut-off of 1.1 nm, and van der Waals interactions were terminated at 1.1 nm with the Verlet cut-off scheme. The simulations were performed with the GROMACS 2020.1 simulation package[73] and conducted analyses with plumed 2.5.3[74]. VMD 1.9.3 was used for visualization purposes.

To quantify the extent of LLPS, we calculated a commonly used description for a contact order parameter [$Q(t)$] defined below:

$$Q(t) = \sum_{i,j} q_{ij}(t), \text{ and}$$

$$q_{ij}(t) = \frac{1 - \left[ r_{ij}(t)/r_0 \right]^6}{1 - \left[ r_{ij}(t)/r_0 \right]^{12}}$$

where $r_{ij}(t)$ is the distance between the $i$-th and $j$-th beads at time $t$ and $r_O$ is fixed to be 0.5 nm. Therefore, $q_{ij}(t)$ can range smoothly from 1 to 0 for a given pair. We calculated the gain/loss of contact ($\Delta Q$) as follows:

$$\Delta Q(t) = Q(t) - N \left\langle q_{single} \right\rangle$$

here, $N$ is the number of IDRs present in the condensate and $\left\langle q_{single} \right\rangle$ is the time-average number of contacts for an isolated IDR.

For the all-atom simulations, we back-mapped the equilibrium CG configuration into atomistic resolution. We used the CHARMM36 force field[75] with the TIP3P water model. We propagated the system for 400 ns (leap-frog integrator with dt = 2 fs) in an NpT ensemble ($T = 298\,K$ and $p = 1\,bar$) with GROMACS 2020.1 MD package. The short-range cut-off for Coulomb and van der Waals interactions was set to 1.2 nm, beyond which the long-range electrostatics were taken care by the Particle Mesh Ewald (PME) method with a Fourier grid spacing of 0.12 nm. All bonds in the same chain were constrained using the LINCS algorithm. We used the Verlet cut-off scheme with a neighbor list update frequency of 40 fs. We monitor $Q(t)$ with time to ensure the equilibration of the condensate and found that $Q(t)$ stabilized after 200 ns. The HBs between different pairs were detected by the geometric criteria, that is, the distance between the donor (D) and acceptor (A) is less than 0.35 nm and the A-D-H angle is less than 30°.

Supplementary Data 1–6 include two final coordinate files of MD simulation for each of the isolated juxtamembrane linker of syt1 (WT, residues 80–142) under normal ionic strength conditions (Supplementary Data 1, 2), the isolated and mutated juxtamembrane linker (residues 80 JuxtaK-142) under normal ionic strength conditions (Supplementary Data 3, 4), and the isolated juxtamembrane linker of syt1 (WT, residues 80–142) under high ionic strength conditions (Supplementary Data 5, 6). Supplementary Data 7–9 includes the MARTINI v3.0 force-field parameters, and Supplementary Data 10–12 includes the GROMACS input files for energy minimization, equilibration, and production runs.

### In vitro droplet formation assay

GFP-fused proteins (in Fig. 3a, b) were assessed for droplet formation under the indicated conditions. Constructs in Fig. 3a contained a flexible GS(GSS)$_4$ segment between the GFP moiety and the indicated syt1 fragments, while constructs in Fig. 3b had no linker. Protein droplets were imaged using a Zeiss 880 Airyscan LSM microscope with a

63X/1.4 NA oil objective at room temperature. Buffer compositions were 25 mM Tris-HCl pH 7.4, 100 mM NaCl and an indicated amount of PEG 8000. Ten µl of the solution containing protein/droplets were placed on 18 mm coverslips (Warner instruments, 64-0734, CS-18R17) and the settled droplets were imaged. For Fig. 5, a Zeiss Axio Vert.AX10 microscope was used to increase imaging throughput. All experiments were done in triplicate, examining more than three fields of view in each trial. In Fig. 5b, the buffers were: 25 mM sodium phosphate (pH 5), 25 mM MES (pH 6), 25 mM HEPES (pH 7), and 25 mM Tris (pH 8, 9), with each solution also containing 100 mM NaCl and 3% PEG 8000. In Fig. 5c, 1,2-dihexanoyl-sn-glycero-3-phospho-L-serine (6:0 PS) was used. In Fig. 5d, $Ca^{2+}$ was buffered using EGTA.

Images were analyzed using Fiji. Briefly, we subjected the image to auto-threshold to create a mask. Using 'Analyze Particles', we determined the size, fluorescence intensity, and number of droplets. Analyzed data were plotted using GraphPad Prism.

## Dynamic Light Scattering (DLS)
DLS was carried out using a DynaPro Nanostar II Dynamic Light Scattering instrument (Waters Wyatt Technology). Protein solutions (10 µM) were buffered in 25 mM Tris pH 7.4, 3% PEG 8000 and indicated NaCl. Average diameter distributions were modeled using Rayleigh Spheres in the DYNAMICS v8 (Waters Wyatt Technology) software. We tested each sample in triplicates, and the results are presented as mean ± SEM.

## Fluorescence recovery after photobleaching (FRAP)
FRAP on protein droplets was carried out using the photobleaching and time series modules of a Zeiss 880 Airyscan LSM microscope with a 63X/1.4 NA oil objective, using Fast Airyscan mode at room temperature. Briefly, we bleached circular regions of interest (1.6 µm in diameter) within 6–10 protein droplets (2–2.5 µm in diameter) per field of view at 70% laser power (488 nm). We performed the imaging at a frame rate of 15 frames per minute. Samples were monitored for 60 s, 500 ms, and 12 min during pre-bleaching, bleaching, and recovery, respectively. All images were processed with automatic Airyscan deconvolution settings. We normalized the fluorescence traces using the equation:

$$FRAP(t) = \frac{F.I._{bleach}(t) - F.I._{background}(t)}{F.I._{non-bleached}(t) - F.I._{background}(t)}$$

where, *F.I.* indicates fluorescence intensity. We performed six FRAP experiments and averaged the *F.I.* data to obtain a single FRAP curve. We propagated errors using pooled variance by assigning weights according to Bessel's correction as $(n_i-1)$, where $n_i$ represents the number of bleached ROIs. Data are represented as mean ± SEM in Fig. 4c.

## Cell culture, transfection, and imaging
HEK293T cells (ATCC, CRL-11268) were maintained in Dulbecco's Modified Eagle Medium (DMEM), high glucose (Gibco, 11965092), supplemented with 10% fetal bovine serum (FBS; R&D Systems, S11550H) and penicillin-streptomycin (Thermo Fisher Scientific, MT-30-001 CI). We dissected and cultured rat hippocampal neurons from pre-natal Sprague-Dawley rats (Envigo) on E18 on P0-P1, as previously described[76]. HEK293T cells and rat neurons were plated on 18 mm coverslips that had been coated with poly-D-lysine (Thermo Fisher Scientific, ICN10269491) for 1 h at room temperature, at a density of 100,000 (HEK293T cells) or 125,000 (neuronal cells) per coverslip, in supplemented DMEM. For neuronal cultures, DMEM was exchanged for Neurobasal-A Media (NBM) (Thermo Fisher Scientific, 10888-022) supplemented with B-27 Media Supplement (Gibco, 17504001), Glutamax (2 mM Gibco, 35050061),

and penicillin-streptomycin, after the neurons had settled (<1 h). Additional supplemented NBM was added every 3–4 days to maintain the health of the neuronal cultures.

Constructs in Fig. 3a were subcloned into pFUGW plasmid for transfection purposes (FUGW was a gift from David Baltimore [Addgene plasmid #14883; http://n2t.net/addgene:14883; RRID: Addgene_14883])[77]. Cells were grown in 12-well culture plates (Genesee Scientific; 25-106); HEK293T cells were transfected 1 day after splitting, while neurons were transfected at 9 day in vitro (DIV), using Lipofectamine LTX reagent with PLUS Reagent (Thermo Fisher Scientific, 15338-100). Briefly, DNA plasmids were diluted in 25 µl Opti-MEM I Reduced Serum Medium (Gibco; 31985062), then 0.25 µl PLUS reagent was added. Separately, 1 µl LTX Reagent was diluted in 25 µl of Opti-MEM I. The DNA-PLUS reagent mixture was added dropwise to the LTX reagent mixture and then to culture media in each well. HEK293T cells and cultured rat hippocampal neurons were imaged in standard extracellular fluid (ECF) imaging solution (140 mM NaCl, 5 mM KCl, 2 mM $CaCl_2$, 2 mM $MgCl_2$, 5.5 mM glucose, 20 mM HEPES (pH 7.3) in PBS) at 37 °C and 5% $CO_2$. All images were processed with automatic Airyscan deconvolution settings. Temperature, $CO_2$, and humidity were controlled using an Oko-lab incubation system. To increase the $[Ca^{2+}]_i$ in HEK293T cells and cultured rat hippocampal neurons, cells were incubated with 1 µM calcimycin (also called as A23187; Sigma, C7522) for 30 min in ECF supplemented with 5 mM $CaCl_2$. To test for reversible dissolution of syt1 droplets, HEK293T cells and cultured rat hippocampal neurons overexpressing syt1 C2AB (80–421)-msGFP forming droplets were treated with 5% of 1,6-hexanediol (Sigma, 88571) for 10 min.

## Protein immunoblots
We collected HEK293T and neuronal cell lysates by harvesting the cultures in 150 µl lysis buffer (2% SDS, 1% Triton X-100, and 10 mM EDTA in PBS) supplemented with a protease inhibitor cocktail (PIC). Samples were boiled at 100 °C for 5 min after addition of 50 µl of 4x Laemmli sample buffer (BioRad, 1610747) containing 2-Mercaptoethanol (BME). Thirty µl of the boiled lysates were resolved on 13% SDS-PAGE gels and subjected to immunoblot analysis as described[76]. Blots were probed in primary antibody (anti-GFP, mAb (clones 7.1 and 13.1), Roche (11814460001), 1:500; anti-syt1, mAb 48 (asv 48), Developmental Studies Hybridoma Bank, 1:500; anti-β-actin, mAb 3700 (8H10D10), Cell Signaling Technology, 1:500), diluted in 2.5% milk in TBST, overnight at 4 °C. Blots were washed thrice and incubated with a secondary antibody (Goat anti-Mouse IgG-HRP, 1706516, Bio-Rad Laboratories, 1:10 K), also diluted in 2.5% milk in TBST, for 1 h, then washed three times for a total of 15 min with TBST. Immunoblots were imaged using Luminata Forte Western HRP substrate (EMD Millipore; ELLUF0100) and a ChemiDoc MP Imaging System (Bio-Rad Laboratories). Bands were analyzed by densitometry, and contrast was linearly adjusted for publication using Fiji.

## Immunocytochemistry
Dissociated rat hippocampal neuronal cultures were fixed with 4% paraformaldehyde, permeabilized with 0.2% saponin, blocked with 0.04% saponin, 10% goat serum, and 1% BSA in PBS, followed by immunostaining (anti-syt1, 105 103, Synaptic Systems, 1:500; anti-syp, 101 004, Synaptic Systems, 1:500) at 4 °C overnight. Cover slips were washed with PBS three times and stained with secondary antibodies (goat anti-rabbit IgG Alexa Fluor 594, 1:1000, Thermo Fisher Scientific (A11037); goat anti-guinea pig IgG Alexa Fluor 647, 1:1000, Thermo Fisher Scientific (A21450)) in 0.1% BSA and 0.04% saponin in PBS for 1 h. Following three more PBS washes, the coverslips were mounted on microscope slides (Thermo Fisher Scientific, 22-178277), using Pro-Long Glass Antifade with Mountant with NucBlue Stain (Thermo Fisher Scientific, P36981), and imaged.

## Statistics and reproducibility

Exact values from experiments and analyses, including the number of trials, are included in the figures or are listed in the Figure Legends. Analyses were performed using GraphPad Prism 9.20 (GraphPad Software Inc). Normality was assessed by histograms of data and QQ plots; if normal, parametric statistical methods were used, if not, nonparametric methods were used for analysis. For all figures, $*p \leq 0.05$, $**p \leq 0.01$, $***p \leq 0.001$, $****p \leq 0.0001$.

## Reporting summary

Further information on research design is available in the Nature Portfolio Reporting Summary linked to this article.

## Data availability

Source data are provided with this paper. Uniprot (syt1 accession IDs: P21707 (Organism: *Rattus norvegicus*), P46096 (Organism: *Mus musculus*), P21579 (Organism: Homo sapiens), P47191 (Organism: *Gallus gallus*), P48018 (Organism: *Bos taurus*), P21521 (Organism: *Drosophila melanogaster*)) and Research Collaboratory for Structural Bioinformatics (RCSB) Protein Data Bank (PDB) (syt1 PDB IDs: 1RSY, 1K5W) databases were used for sequence and structure analysis. Source data are provided with this paper.

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

## Acknowledgements

We thank all members of the Chapman lab for helpful discussions. This work was funded by the National Institutes of Health (grants MH061876 and NS097362 to E.R.C). The computational component is supported by a grant from the National Science Foundation (CHE-2154804) to Q.C. Computational resources from the Extreme Science and Engineering Discovery Environment (XSEDE), which is supported by NSF Grant ACI-1548562, are greatly appreciated; part of the computational work was performed on the Shared Computing Cluster, which is administered by Boston University's Research Computing Services (www.bu.edu/tech/

support/research/). E.R.C. is an Investigator of the Howard Hughes Medical Institute (HHMI). This article is subject to HHMI's Open Access to Publications policy. HHMI lab heads have previously granted a non-exclusive CC BY 4.0 license to the public and a sublicensable license to HHMI in their research articles. Pursuant to those licenses, the author-accepted manuscript of this article can be made freely available under a CC BY 4.0 license immediately upon publication.

## Author contributions

N.M., Q.C. and E.R.C. designed the study; N.M., S.M. and E.T.W. performed experiments and analyzed data; N.M. and E.R.C. wrote the manuscript; E.R.C. and Q.C. provided funding.

## Competing interests

The authors declare no competing interests.

## Ethics approval

Animal care and use in this study were conducted under guidelines set by the National Institutes of Health's *Guide for the care and use of laboratory animals* handbook. Protocols were reviewed and approved by the Animal Care and Use Committee at the University of Wisconsin–Madison (Laboratory Animal Welfare Public Health Service Assurance Number: A3368-01).
