## [Peer Review File · Nature Communications]

The juxtamembrane linker of synaptotagmin 1 regulates Ca²⁺ binding via liquid-liquid phase separationREVIEWER COMMENTS

Reviewer #1 (Remarks to the Author):

Mehta et al. have conducted an interesting study to examine the role of the syt1 juxtamembrane linker in liquid-liquid phase separation (LLPS). The authors make a compelling case that the lysine-rich sequence of the linker facilitates LLPS. The combination of experiments and simulations is a potential strength of this work. However, I have concerns about the conclusiveness of the findings that should be addressed before the work is suitable for publication in Nature Communications.

The title and stated implications regarding Ca²⁺ binding regulation seem premature given the current scope of the study. I would recommend tempering the conclusions and more clearly linking the results to proposed functional regulation of Ca²⁺ interactions. Additional experiments focused on Ca²⁺ effects could strengthen these connections.

Some specific suggestions to enhance the rigor and impact:

- 1) The observation of stable lysine-glutamate/aspartate interactions highlights the likely importance of electrostatic interactions. Systematically analyzing charge composition (e.g. total charge fraction, charge asymmetry) and modulation of LLPS propensity could provide greater mechanistic insight into the role of electrostatics.
- 2) Exploring lysine to arginine mutations computationally and experimentally is an intriguing idea that could reveal impacts on phase separation beyond just charge.
- 3) Comparing disorder predictions to recent AlphaFold2 pLDDT scores would utilize state-of-the-art computational methods to evaluate intrinsic disorder.
- 4) Providing full details on all Martini3 simulation parameters and ideally sharing input files publicly would significantly aid reproducibility of the modeling.
- 5) Disclosing the number of independent repeats for each model (at least two for better statistics) and adding representative trajectories/videos could help substantiate the conclusions drawn from modeling.

Additional aspects that could further extend the impact:

6) Performing Martini3 simulations on syt1 C2AB domains could place the linker in its wider protein context.

7) Simulating the linker with Ca²⁺ could offer more physiologically relevant insights into regulation.

In summary, I see this as an interesting starting point on the syt1 linker. Expanding the rigor and scope of the study as outlined above could produce a more definitive work suitable for Nature Communications.

Reviewer #2 (Remarks to the Author):

Summary

This manuscript identifies the functional role of the juxtamembrane linker of synaptotagmin1 in its Ca²⁺ binding through liquid-liquid phase separation. Using a SUMO tag, the authors purified the entire cytoplasmic domain of synaptotagmin1. Surprisingly, the linker reduces the affinity of C2A for calcium. Additionally, they found that the IDR region of synaptotagmin1 mediates liquid-liquid phase separation of this protein, not only in vitro but also in cells. This manuscript addresses an interesting topic, and overall, the results are important. However, some concerns need to be addressed for further improvement.

Major comments.

1. Some previous studies have shown that EGFP enhances liquid-liquid phase separation or aggregation of certain proteins. Therefore, the authors should consider testing the liquid-liquid phase separation of this protein with different tag both in vitro and in vivo.

2. Previous studies have demonstrated that synaptotagmin1 full-length does not form droplets but primarily localizes on the plasma membrane when expressed in non-neuronal cells. How did the authors rule out the potential artifacts of aggregation caused by a truncated form of proteins?

3. The author found that synaptotagmin1 C2AB (80-421)-EGFP forms droplets in HEK cells

and cultured neurons. What happens if the authors increase the intracellular Ca^{2+} level in cells by using some ionophores?

4. Can synaptotagmin1 C2AB (80-421)-EGFP condensates in both HEK cells and cultured neurons be reversibly dispersed by 1,6-Hexanediol?

5. What is the expression pattern of the synaptotagmin1 C2AB (80-421)-EGFP construct in axons of neurons, and does it co-localize or co-oligomerize with wild-type synaptotagmin1 on synaptic vesicles?

Minor comments

1. This journal caters to a diverse audience with varying knowledge and backgrounds. Some readers may encounter difficulties in understanding certain experiments in this manuscript, such as Figure 1c-f. It would be greatly appreciated if the authors could include additional details and experimental procedures.

(Black- Reviewer comments, Blue- Response to reviewers, Green- Added in the manuscript)

Reviewer #1:

Mehta et al. have conducted an interesting study to examine the role of the syt1 juxtamembrane linker in liquid-liquid phase separation (LLPS). The authors make a compelling case that the lysine-rich sequence of the linker facilitates LLPS. The combination of experiments and simulations is a potential strength of this work. However, I have concerns about the conclusiveness of the findings that should be addressed before the work is suitable for publication in Nature Communications.

The title and stated implications regarding Ca^{2+} binding regulation seem premature given the current scope of the study. I would recommend tempering the conclusions and more clearly linking the results to proposed functional regulation of Ca^{2+} interactions. Additional experiments focused on Ca^{2+} effects could strengthen these connections.

We thank the reviewer for their summary and insightful comments. We address each of the questions that were raised, below.

Some specific suggestions to enhance the rigor and impact:

1) The observation of stable lysine-glutamate/aspartate interactions highlights the likely importance of electrostatic interactions. Systematically analyzing charge composition (e.g. total charge fraction, charge asymmetry) and modulation of LLPS propensity could provide greater mechanistic insight into the role of electrostatics.

We agree with the reviewer that further characterization of the charge composition of the syt1 intrinsically disordered region (IDR) (residues 80-142) would provide greater mechanistic insights. To address this issue, we have calculated several parameters generally used to characterize charged IDR segments, namely, (i) the fraction of charged residues (FCR), (ii) net charge per residue (NCPR), and (iv) sequence charge decoration (SCD). The results are summarized in Supplementary Table 2 below:

Parameters	Expression	Value
Fraction of positive charges (f_+)	N_+/N	0.301
Fraction of negative charges (f_-)	N_-/N	0.254
FCR	$(f_+ + f_-)$	0.555
NCPR	$ f_+ - f_- $	0.047
SCD	$\frac{1}{N} \left[\sum_{m=2}^N \sum_{n=1}^{m-1} q_m q_n (m-n)^{1/2} \right]$	-21.5

Supplementary Table 2

Charge composition of the syt1 IDR

Definitions:

- a) f_+ (fraction of positive charges): ratio of the number of positive charges (N_+) to total number of residues (N).
- b) f_- (fraction of negative charges): ratio of the number of negative charges (N_-) to total number of residues (N).
- c) FCR (fraction of charged residues): sum of the fraction of positive (f_+) and negative charges (f_-).
- d) NCPR (net charge per residue): difference between the fraction of positive (f_+) and negative charges (f_-).
- e) SCD (sequence charge decoration): SCD quantifies charge patterning in a sequence, indicated by the expression in the table. It takes into account the charges (q_m and q_n), along with their separation within the primary protein sequence ($m-n$).

Previous studies have analyzed the correlation between these charge composition properties with the propensity of proteins to undergo LLPS. An FCR value greater than 0.3, or a high negative value of the SCD parameter, indicates a ‘strong’ polyampholyte. It was shown in ref. 1-5 (given below)¹⁻⁵, as well as in additional studies, that the charge blockiness (related to a high negative SCD and a high FCR value) increases the window of phase separation and makes the LLPS droplet more resilient to increasing salt concentrations.

Therefore, the IDR sequence can be characterized as a charge-rich, non-neutral, and strong polyampholyte-like segment. We added the following description in the **Results** section of the revised manuscript:

...To further address the role of charged residues in LLPS of syt1, we determined the fraction of positive and negative charges, fraction of charged residues (FCR), net charge per residue (NCPR), and sequence charge decoration (SCD) values for the juxtamembrane linker (residues 80-142) were calculated as shown in Supplementary Table 2. We obtained an FCR value (0.555) greater than 0.3, and a high negative value of the SCD parameter (-21.5); both are predictive of a ‘strong’ polyampholyte that is likely to undergo phase separation. These values also suggest the condensates are resilient to increasing salt concentrations^{50,51}. These findings further imply a crucial role of the lysine residues in the juxtamembrane linker of syt1 to facilitate LLPS.

2) Exploring lysine to arginine mutations computationally and experimentally is an intriguing idea that could reveal impacts on phase separation beyond just charge.

We note that there exist detailed analyses in the literature regarding the role of lysine (K) and arginine (R) residues in the LLPS propensities of IDR sequences^{6,7}. In general, it was observed that arginine-rich proteins undergo LLPS more readily than lysine-rich ones. Arginine-rich sequences are also found to exhibit reentrant LLPS behavior⁸. For example, in the context of LAF-1 RGG domain, Mittal and co-workers⁷ showed that R to K mutations reduced the propensity to undergo LLPS due to the loss of sp^2/π and cation- π interactions. In the context of protein/RNA LLPS, Barducci *et al.*⁶ found that arginine possesses an increased affinity (compared to lysine) for poly-uracil even under dilute conditions, due to the tendency of arginine sidechains to form a higher number of specific interactions (hydrogen bond and π -stacking) with polynucleotides. It is likely that the same kinds of effects would be observed regarding the lysine residues of syt1; that is, substitution of the lysine with arginine residues of the linker would enhance LLPS. We feel this is a rather subtle point so have opted to focus our revision efforts on other issues raised by the

referees. We agree that further structure-function experiments, including the lysine-to-arginine mutations, should be pursued in a future study.

3) Comparing disorder predictions to recent AlphaFold2 pLDDT scores would utilize state-of-the-art computational methods to evaluate intrinsic disorder.

This is a good point. To address this, we obtained the predicted AlphaFold^{9,10} structure of syt1 from the UniProtKB database (P21707) and now show this, along with predicted local distance difference test (pLDDT) per-residue confidence scores, in new Supplementary Fig. 2. According to the AlphaFold prediction, the region of residues 80-142 is mostly unstructured with a pLDDT score less than 50; there is a short sequence that is predicted to be helical, with a higher pLDDT score, as indicated in the figure below:

Supplementary Fig. 2. Color-coded AlphaFold predicted syt1 structure, along with pLDDT scores, illustrating both folded and unstructured domains.

a AlphaFold predicts that the juxtamembrane linker segment, residues Lys80 to Leu142, is mostly unstructured with a pLDDT score less than 50. There is a short sequence in the juxtamembrane linker segment that is predicted to be helical, with a higher pLDDT score. pLDDT: predicted local distance difference test per-residue confidence score.

We have included the following description in the Results section of the manuscript:

...These structural predictions were further confirmed using AlphaFold^{9,10} (Supplementary Fig. 2).

4) Providing full details on all Martini3 simulation parameters and ideally sharing input files publicly would significantly aid reproducibility of the modeling.

We apologize for this oversight; during revision we provided the simulation parameters used in GROMACS in three *.itp and *.mdp files, as a Supplementary Dataset.

Supplementary Dataset:

- 1) Three *.itp files: Martini3 force field parameters files.

2) Three *.mdp files: GROMACS input files for energy minimization, equilibration, and production run.

5) Disclosing the number of independent repeats for each model (at least two for better statistics) and adding representative trajectories/videos could help substantiate the conclusions drawn from modeling.

This is a valid point, we thank the reviewer for the suggestion. Previously, we reported observations from one MD trajectory per system. During revision, we repeated these efforts and obtained an additional, independent, 5 μ s trajectory for each system. We have also provided the relevant simulations as Supplementary Movies 1-3, and 6 and in the Supplementary Dataset section of the revised manuscript, as shown below:

Supplementary Movie 1. MD simulation shows that the isolated juxtamembrane linker of syt1 stably self-associates. Box edge, 30 nm.

Supplementary Movie 2. MD simulation shows that the isolated, and mutated juxtamembrane linker of syt1 (residues 80 JuxtaK-142) dissolves within 5 μ s of simulation. Box edge, 30 nm.

Supplementary Movie 3. MD simulation shows that the complete cytoplasmic domain of syt1 (residues 80-421) stably self-associates. Box edge, 35 nm.

Supplementary Movie 6. MD simulation shows that the droplet formed by the isolated juxtamembrane linker of syt1 molecules dissolves at high ionic strength (400 mM NaCl). Box edge, 30 nm.

Supplementary Dataset:

1) Six *.gro files: two MD simulations final coordinate files for each of the (a) isolated juxtamembrane linker of syt1 (residues 80-142), (b) isolated and mutated juxtamembrane linker (residues 80 JuxtaK-142), and (c) isolated juxtamembrane linker of syt1 (residues 80-142) under high ionic strength condition.

Additional aspects that could further extend the impact:

6) Performing Martini3 simulations on syt1 C2AB domains could place the linker in its wider protein context.

We thank the reviewer for this useful suggestion. To address this, we have performed additional simulations of two systems: (i) The full sequence (residues 80-421) and (ii) the C2AB domains (without the IDR; residues 143-421). These simulations were performed with the same scaled MARTINI v3.0 force field parameters and the LLPS findings are provided in a new Supplementary Fig. 3. It is clear that the presence of the linker drives LLPS. A low level of clustering was observed for syt1 C2-domains without the juxtamembrane linker, but this involved only a small number of molecules.

Supplementary Fig. 3. The syt1 juxtamembrane linker drives LLPS.

a Probability distribution of the number of syt1 molecules, with and without the juxtamembrane linker, in the largest cluster observed along the MD trajectories. Syt1 molecules with the juxtamembrane linker tend to form larger clusters, indicating that the presence of linker drives LLPS. **b** Probability distribution of the number of syt1 clusters (N_{clust}), with and without the juxtamembrane linker. Removal of the linker results in the formation of a larger number of smaller clusters. **c** Illustration of syt1 clusters; a small number of large clusters form with an intact juxtamembrane linker (upper); a larger number of smaller clusters form in the absence of the linker (lower).

We have included the above findings in the Results section of our revised manuscript:

...Next, we performed coarse-grained MD simulations of the cytoplasmic domain of syt1, with and without the juxtamembrane linker, to further address the role of this segment in LLPS. As shown in Supplementary Fig. 3a-c, syt1 molecules, with an intact juxtamembrane linker, have a higher probability of forming large clusters, while syt1 fragments lacking the juxtamembrane linker formed a larger number of smaller clusters.

The MD simulation protocol has also been added in the Methods section, as shown below:

...Additionally, MD simulations involving the full cytoplasmic domain (residues 80-421) and the C2AB domain (residues 143-421, without the linker) of syt1 were performed with the same force field parameters. For each of these simulations, we solvated 20 protein molecules within a 35 nm cubic box.

7) Simulating the linker with Ca^{2+} could offer more physiologically relevant insights into regulation. We appreciate the referee's suggestion. However, there is no evidence that Ca^{2+} ions interact with the linker region; this region does not contain any putative Ca^{2+} binding motifs. Moreover, for completeness, we directly measured Ca^{2+} -binding to this segment via isothermal titration calorimetry (ITC), and no significant binding was detected (unpublished observations).

Our hypothesis is that the linker drives LLPS which in turn partially masks/hides the Ca^{2+} binding sites in the C2-domains of syt1. To test this, we computed the solvent accessible surface area (SASA) values of the ten aspartate residues that mediate Ca^{2+} binding in each C2-domain, in syt1 condensates versus the isolated protein. We found that the SASA values of the aspartate residues within the condensate is reduced significantly for six out of ten aspartate residues (Supplementary Fig. 13). These findings are consistent with the idea that the Ca^{2+} binding sites are hidden from

the environment due to LLPS driven by the juxtamembrane linker region (please also see our response to the previous point - #6 – as well as Supplementary Fig. 3).

Supplementary Fig. 13. LLPS masks Ca^{2+} -binding to the aspartate residues in the C2-domains of syt1.

Solvent accessible surface area (SASA) values for the ten aspartate residues proposed to mediate Ca^{2+} binding in the tandem C2-domains of syt1. The green squares denote the SASA values for a single isolated protein and the hollow obliques denote the SASA values for the same aspartate residues in twenty different copies of the protein in a condensate. Two sided unpaired t-tests were performed with three datasets using the two-stage step-up method of Benjamini, Krieger, and Yekutieli¹¹; p-values of 0.007, 0.006, 0.013, 0.005, 0.012, and 0.03 were obtained for residues D172, D178, D230, D309, D363, and D365 respectively.

These new findings have been added to Results section of our revised manuscript.

...In summary, we return to our findings above, that provide new insights regarding the interaction of syt1 with Ca^{2+} . Namely, the juxtamembrane linker reduces the affinity of the C2-domains of syt1 for Ca^{2+} (Fig. 1c-f), Ca^{2+} enhances syt1 LLPS (Fig. 5d), and increased $[\text{Ca}^{2+}]_i$ facilitates the fusion of syt1 droplets in cells (Fig. 7). We therefore examined the ten aspartate residues that coordinate Ca^{2+} ions in the tandem C2-domains of syt1. We assessed the solvent accessible surface area (SASA) values of these residues in a syt1 condensate, as compared to the isolated protein. Six out of the ten aspartate side chains had significantly lower SASA values, resulting in the partial masking of these Ca^{2+} -binding residues within droplets (Supplementary Fig. 13). These findings explain how the linker reduces the apparent affinity of syt1 for Ca^{2+} ; this is an indirect effect mediated by LLPS. How Ca^{2+} facilitates droplet formation and fusion between droplets, remain to be established.

In summary, I see this as an interesting starting point on the syt1 linker. Expanding the rigor and scope of the study as outlined above could produce a more definitive work suitable for Nature Communications.

Reviewer #2:

Summary

This manuscript identifies the functional role of the juxtamembrane linker of synaptotagmin1 in its Ca²⁺ binding through liquid-liquid phase separation. Using a SUMO tag, the authors purified the entire cytoplasmic domain of synaptotagmin1. Surprisingly, the linker reduces the affinity of C2A for calcium. Additionally, they found that the IDR region of synaptotagmin1 mediates liquid-liquid phase separation of this protein, not only *in vitro* but also in cells. This manuscript addresses an interesting topic, and overall, the results are important. However, some concerns need to be addressed for further improvement.

We thank the reviewer for this summary, and we appreciate the positive feedback. We have addressed each of the questions that were raised, below.

Major comments.

1. Some previous studies have shown that EGFP enhances liquid-liquid phase separation or aggregation of certain proteins. Therefore, the authors should consider testing the liquid-liquid phase separation of this protein with different tag both *in vitro* and *in vivo*.

The reviewer raises an important point: EGFP enhances LLPS, or aggregation, when appended to certain proteins. We are very much aware of this issue, so in our study we were careful to use a mutated version of EGFP, called msGFP (i.e., **monomeric superfolder GFP**)¹², that does not aggregate and expresses as a stable monomer^{13,14}. It has also been reported that msGFP does not cause proteins, that it is appended to, to phase separate or to aggregate^{12,15}. Second, all experiments our study include a negative control - msGFP alone – showing this tag does not aggregate or undergo LLPS (Fig. 3c,d, Fig. 6, Supplementary Fig. 5 and 6). Third, to directly address the reviewer's concern we now include new experiments in our revised manuscript, where we examined potential droplet formation, using complete and truncated cytoplasmic domains of syt1 C2AB, but now labeled with an organic dye, fluorescein, rather than a GFP variant. These experiments were carried out under the same buffer conditions used for the msGFP fusion proteins in the original manuscript, including the presence of 3% PEG 8000. As expected, and shown in a new **Supplementary Fig. 5**, the fluorescein-labeled complete cytoplasmic domain of syt1 (residues 80-421) formed droplets, whereas no droplet formation was observed using the fluorescein-labeled truncated cytoplasmic domain of syt1 (residues 143-421).

We reiterate and emphasize that msGFP has been used as a fluorescent tag to assay for droplet formation in a number of cell-based studies¹⁶⁻¹⁹. Thus, using this monomeric form of GFP, the inclusion of negative controls, along with our new findings using organic dye-labeled protein, we conclude that msGFP does not influence liquid-liquid phase separation of the cytoplasmic domain of syt1 *in vitro* or in cell-based experiments. During revision, we added a new **Supplementary Fig. 5**, showing the completed cytoplasmic domain of syt1, bearing a fluorescein-label, forms droplets, whereas the truncated cytoplasmic domain, lacking the linker and IDR, does not.

Supplementary Fig. 5. Fluorescein labeled syt1 forms droplets via the juxtamembrane linker segment.

a (Left) Syt1 C2AB (80-421), labeled with an organic dye (fluorescein) at residues Cys82 and Cys277, formed droplets in the presence of 3% PEG 8000. The buffer condition was 25 mM Tris-HCl (pH 7.4), 100 mM NaCl. The truncated cytoplasmic domain of syt1 (residues 143-421; right), lacking the juxtamembrane linker (residues 80-142) failed to form droplets. Scale bars, 3 μm.

We include our new Supplementary Fig. 5 and add the text below in the Results and Discussion section.

...We emphasize that we used a monomeric, superfolder version of GFP (see Methods) that does not promote aggregation when fused to other proteins^{13,14}. This version of GFP has been used to study LLPS in numerous studies¹⁶⁻¹⁹, and was used, in an untagged form, as a negative control in our studies.

...To confirm these results, we labeled both the complete and truncated cytoplasmic domain of syt1 (residues 80-421 and 143-421, respectively) with an organic dye (fluorescein) at native Cys residues 82 and 277. As expected, we observed droplet formation with the complete cytoplasmic domain of syt1 but not for the truncated protein lacking the linker (Supplementary Fig. 5).

The dye labeling protocol has also been added in the Methods section, as shown below:

...For organic dye-labeled experiments, syt1 cytoplasmic domain (either residues 80-421 or 143-421) was labeled with fluorescein at Cys82 and Cys277. Twenty-five μM syt1 protein was incubated with a 10-fold excess of fluorescein-5-maleimide overnight at 4°C. Free dye was removed using a PD-10 desalting column.

2. Previous studies have demonstrated that synaptotagmin1 full-length does not form droplets but primarily localizes on the plasma membrane when expressed in non-neuronal cells. How did the authors rule out the potential artifacts of aggregation caused by a truncated form of proteins?

This is an interesting point. We view this question in light of our atomic force microscopy (AFM) findings in Courtney et al. (2021)²⁰, where, under native conditions (i.e., in aqueous buffer over lipid bilayers), the intact cytoplasmic domain of syt1 (80-421) formed large ring-like structures, and patches, on the bilayer²⁰. We view these irregular structures as LLPS phase separated protein on the 2D surface of the bilayers. Syt1 cannot form droplets in the cytosol, as it is an integral membrane protein. We clarify this point in the Discussion section of our revised manuscript by stating:

...In light of the discovery that syt1 undergoes LLPS, we note atomic force microscopy (AFM) images of the intact cytoplasmic domain of syt1, under aqueous conditions on lipid bilayers, revealed the formation of large ring-like structures and protein patches (Courtney et al. (2021)²⁰. We propose that these structures reflect syt1 LLPS on the two-dimension surface of membranes.

3. The author found that synaptotagmin1 C2AB (80-421)-EGFP forms droplets in HEK cells and cultured neurons. What happens if the authors increase the intracellular Ca²⁺ level in cells by using some ionophores?

This is an excellent idea. We thus performed experiments using a Ca²⁺ ionophore, calcimycin (or A23187) in HEK293T cells and cultured rat hippocampal neurons overexpressing syt1 C2AB (80-421)-msGFP. Interestingly, we observed that the number of droplets decreased, due - at least in part - to fusion between droplets, thus increasing the area of the droplets. Droplet growth might also occur due to increased partitioning of protein into the LLPS droplets, but at present we do not have a means to discern the relative contributions of these two potential means of droplet growth. Regardless, the ability of Ca²⁺ to drive fusion between droplets is a new observation that increases the novelty of our study. The effect of Ca²⁺ is illustrated in a new figure (Fig. 7), added to our study during revision, as shown below:

Fig. 7. Ca²⁺ influx, via an ionophore, drives syt1 droplet fusion in HEK293T cells and cultured rat hippocampal neurons. Left: HEK293T cells; Right: cultured rat hippocampal neurons. **a** Representative images of HEK293T cells/hippocampal neurons overexpressing syt1

C2AB (80-421)-GFP showing protein droplets under control and + calcimycin (a Ca^{2+} ionophore) conditions. Two inset images, shown at higher magnification, reveal the larger droplet size in the + calcimycin condition. **b** Representative line scan traces of dashed lines in **a** (i) are fluorescence intensity of syt1 droplets under control and + calcimycin conditions. **c-e** Plots of syt1 droplet fluorescence intensity (F.I.), number, and area. Calcimycin treatment had no effect on F.I. but decreased the number of droplets and increased their area (Number: 0.73 ± 0.1 SD (HEK293T), 0.70 ± 0.1 SD (neurons); Area: p-values (calcimycin vs control) 0.016 (HEK293T), 0.042 (neurons)). A total of nine fields of view were analyzed, and droplet area was measured as described in Methods. Mean values are indicated by the horizontal line in each panel. In panel **e**, the p-values were calculated using the Mann Whitney two-sided test. Scale bars, 10 μm . Inset scale bars, 1 μm .

We include these findings in a new Fig. 7, along with the added text in Abstract, Results and Discussion in our revised manuscript.

... increases in $[\text{Ca}^{2+}]_i$ promote the fusion of syt1 droplets in living cells.

...Since Ca^{2+} enhanced LLPS of the intact cytoplasmic domain of syt1 *in vitro*, we assessed the effect of a Ca^{2+} ionophore, calcimycin or A23187, on these droplets in cells. Representative images and line traces (Fig. 7a,b) show an increase in syt1 (80-421)-GFP droplet area in the presence of calcimycin (5 mM Ca^{2+} in the ECF) as compared to the control condition in HEK293T (left) and rat hippocampal neurons (right). As described in image analysis (Methods), we carefully assessed the fluorescence intensity, number, and area of syt1 droplets under both conditions for the two cell types. Although the fluorescence intensity of syt1 droplets remained unchanged across the two conditions (Fig. 7c), there was a decrease in the number of syt1 droplets in the presence of calcimycin (Fig. 7d). Quantification of droplets area revealed a significant increase in area of syt1 droplets under calcimycin conditions in both cell types (Fig. 7e), due to fusion between droplets, as detailed below in the Discussion.

...Interestingly, upon mobilization of $[\text{Ca}^{2+}]_i$ with an ionophore, we observed fusion between syt1 condensates, resulting in larger droplets. Droplet growth might also occur due to increased partitioning of protein into the LLPS droplets; at present we are not able discern the relative contributions of these two potential means of droplet growth.

The Methods section detailing the ionophore protocol was updated, as shown below:

...To increase the $[\text{Ca}^{2+}]_i$ in HEK293T cells and cultured rat hippocampal neurons, cells were incubated with 1 μM calcimycin (or A23187, Sigma, C7522) for 30 min in ECF supplemented with 5 mM CaCl_2 .

4. Can synaptotagmin1 C2AB (80-421)-EGFP condensates in both HEK cells and cultured neurons be reversibly dispersed by 1,6-Hexanediol?

The reviewers suggested that we verify a property of liquid-liquid phase separation (LLPS) by assaying for reversible dispersion/dissolution of protein droplets using aliphatic alcohols, such as 1,6-hexanediol. This chemical interferes with weak hydrophobic interactions within a droplet and dissolves the condensate, showcasing a reversible property of LLPS droplets²¹. To address this issue, we performed experiments involving 1,6-hexanediol treatment of HEK293T cells and

cultured rat hippocampal neurons overexpressing syt1 C2AB (80-421)-GFP protein. As expected for liquid-like droplets, syt1 droplets – in both cell types – dissolved (partially or completely) upon treatment with 1,6-hexanediol. These findings are now included in our revised study, in Supplementary Fig. 10, and the experimental approach is detailed in Methods.

Supplementary Fig. 10. 1,6-hexanediol dissolves syt1 droplets in HEK293T cells and cultured rat hippocampal neurons.

a Representative super-resolution fluorescence images of HEK293T cells overexpressing syt1 (80-421)-GFP under control and 1,6-hexanediol treatment (5%) conditions. 1,6-hexanediol treatment for 10 min dissolved syt1 droplets. **b** same as **a**, but in cultured rat hippocampal neurons. Scale bars, 10 μm .

We include these findings in a new Supplementary Fig. 10, along with Results in our revised manuscript.

...To further characterize these droplets, we examined their potential dissolution using 1,6-hexanediol, an aliphatic alcohol that interferes with weak hydrophobic interactions within droplets. Indeed, syt1 droplets dissolved (partially or completely) upon treatment with 1,6-hexanediol in

both HEK293T cells and neurons, as shown by representative images in Supplementary Fig. 10a,b.

The Methods section detailing hexanediol treatment protocol was updated, as shown below:

...To test for reversible dissolution of syt1 droplets, HEK293T cells and cultured rat hippocampal neurons overexpressing syt1 C2AB (80-421)-msGFP forming droplets were treated with 5% of 1,6-hexanediol (Sigma, 88571) for 10 min.

5. What is the expression pattern of the synaptotagmin1 C2AB (80-421)-EGFP construct in axons of neurons, and does it co-localize or co-oligomerize with wild-type synaptotagmin1 on synaptic vesicles?

The reviewer raises an important aspect of expression pattern of syt1 (80-421)-GFP in the axons of neurons and to check for colocalization with WT syt1 protein on synaptic vesicles. To address this, we performed immunocytochemistry on fixed cultured rat hippocampal neurons. We stained endogenous syt1 using an antibody directed against its luminal domain, and we detected syt1 (80-421)-GFP via GFP fluorescence. We stained for synaptophysin (syp), as a canonical SV marker. Representative figures, in a new Supplementary Fig. 11, reveal that syt1 (80-421)-GFP colocalizes, in part, with syp and endogenous syt1, in transfected neurons (Supplementary Fig. 12). These findings are consistent with the idea that syt1 (80-421)-GFP interacts with endogenous syt1, via LLPS.

Supplementary Fig. 11. Partial co-localization of syt1 (80-421)-GFP with synaptic vesicle proteins in cultured hippocampal neurons.

a Localization of syt1 (80-421)-GFP was visualized in neurites, whereas endogenous synaptophysin (syp) and syt1 were visualized via immunocytochemistry (ICC); syt1 was selectively detected using an N-terminal luminal domain antibody. Syt1 (80-421)-GFP partially colocalized with endogenous syt1 (**a**) and syp (**b**) as indicated by the arrows in the magnified insets. **c** endogenous syt1 and syp colocalized to a large extent. In all cases, images were adjusted with linear brightness and contrast. Scale bars, 10 μ m. Inset scale bars, 5 μ m.

Supplementary Fig. 12. Colocalization quantification of syt1 (80-421)-GFP, endogenous syt1, and syp.

a Colocalization analysis yields Mander's overlap colocalization (MOC) of endogenous syt1 and syp as 0.66 ± 0.01 SEM and 0.79 ± 0.02 SEM. **b,c** same as **a**, but for syt1 (80-421)-GFP with endogenous syt1 and syp, yielding MOCs of 0.18 ± 0.01 SEM and 0.20 ± 0.03 SEM respectively.

We include these findings in two new Supplementary Fig. 11,12, along with Results and Discussion in our revised manuscript.

...We also addressed the question of whether syt1 (80-421)-GFP might interact, via LLPS, with native syt1 on synaptic vesicles, by examining the colocalization of overexpressed syt1 (80-421)-GFP with endogenous syt1 and synaptophysin (syp, a canonical synaptic vesicle marker) using immunocytochemistry (ICC). Endogenous syt1 was selectively probed using a N-terminal luminal domain antibody. As indicated by the arrowheads in the magnified inset images of Supplementary Fig. 11a,b, syt1 (80-421)-GFP partially colocalizes with endogenous syt1 and syp (Supplementary Fig. 12b,c); the degree of co-localization was limited, as the over-expressed truncated fusion protein was found throughout the neurons. As expected, the two native synaptic vesicle markers were strongly colocalized (Supplementary Fig. 11c, 12a). We note that all neurons express endogenous syp and syt1, but only a handful of cells expressed the transfected syt1 (80-421)-GFP construct, lacking the N-terminal domain. While these co-localization studies are consistent with LLPS-mediated interactions between recombinant and native syt1, future studies will focus on direct measurements of these interactions, and their impact on SV clustering.

...Furthermore, the overexpressed complete cytoplasmic domain of syt1 partially colocalizes with endogenous syt1 at synapses, potentially via LLPS.

We also updated our **Methods** section, as follows:

Immunocytochemistry

Dissociated rat hippocampal neuronal cultures were fixed with 4% paraformaldehyde, permeabilized with 0.2% saponin, blocked with 0.04% saponin, 10% goat serum, and 1% BSA in PBS, followed by immunostaining (anti-syt1, 105 103CpH, Synaptic Systems, 1:500; anti-syp, 101 004, Synaptic Systems, 1:500) at 4°C overnight. Cover slips were washed with PBS three times and stained with secondary antibodies (goat anti-rabbit IgG Alexa Fluor 594, 1:1000, Thermo Fisher Scientific (A11037); goat anti-guinea pig IgG Alexa Fluor 647, 1:1000, Thermo Fisher Scientific (A21450)) in 0.1% BSA and 0.04% saponin in PBS for 1 hr. Following three more PBS washes, the coverslips were mounted on microscope slides (Thermo Fisher Scientific, 22-178277), using ProLong Glass Antifade with Mountant with NucBlue Stain (Thermo Fisher Scientific, P36981), and imaged.

Statistics

Exact values from experiments and analyses, including the number of trials, are included in the figures or are listed in the Figure Legends. Analyses were performed using GraphPad Prism 9.20 (GraphPad Software Inc). Normality was assessed by histograms of data and QQ plots; if normal, parametric statistical methods were used, if not, nonparametric methods were used for analysis. For all figures, * $p \leq 0.05$, ** $p \leq 0.01$, *** $p \leq 0.001$, **** $p \leq 0.0001$.

Minor comments

1. This journal caters to a diverse audience with varying knowledge and backgrounds. Some readers may encounter difficulties in understanding certain experiments in this manuscript, such as Figure 1c-f. It would be greatly appreciated if the authors could include additional details and experimental procedures.

We thank the reviewer for bringing this crucial point to our attention. We address this concern in our manuscript by adding the following in our **Results** section of revised manuscript:

...In these traces, each injection of Ca^{2+} results in a peak, as heat is either liberated or absorbed. Ca^{2+} injections progress from left to right, with the peak amplitude diminishing as the protein becomes saturated.

References

1. Das, R. K. & Pappu, R. V. Conformations of intrinsically disordered proteins are influenced by linear sequence distributions of oppositely charged residues. *Proc. Natl. Acad. Sci.* **110**, 13392–13397 (2013).
2. Rumyantsev, A. M., Johner, A. & De Pablo, J. J. Sequence Blockiness Controls the Structure of Polyampholyte Necklaces. *ACS Macro Lett.* **10**, 1048–1054 (2021).
3. McCarty, J., Delaney, K. T., Danielsen, S. P. O., Fredrickson, G. H. & Shea, J.-E. Complete Phase Diagram for Liquid–Liquid Phase Separation of Intrinsically Disordered Proteins. *J. Phys. Chem. Lett.* **10**, 1644–1652 (2019).
4. Das, S., Eisen, A., Lin, Y.-H. & Chan, H. S. A Lattice Model of Charge-Pattern-Dependent Polyampholyte Phase Separation. *J. Phys. Chem. B* **122**, 5418–5431 (2018).

5. Lin, Y.-H., Brady, J. P., Chan, H. S. & Ghosh, K. A unified analytical theory of heteropolymers for sequence-specific phase behaviors of polyelectrolytes and polyampholytes. *J. Chem. Phys.* **152**, 045102 (2020).
6. Paloni, M., Bailly, R., Ciandrini, L. & Barducci, A. Unraveling Molecular Interactions in Liquid–Liquid Phase Separation of Disordered Proteins by Atomistic Simulations. *J. Phys. Chem. B* **124**, 9009–9016 (2020).
7. Schuster, B. S. *et al.* Identifying sequence perturbations to an intrinsically disordered protein that determine its phase-separation behavior. *Proc. Natl. Acad. Sci.* **117**, 11421–11431 (2020).
8. Hong, Y. *et al.* Hydrophobicity of arginine leads to reentrant liquid-liquid phase separation behaviors of arginine-rich proteins. *Nat. Commun.* **13**, 7326 (2022).
9. Jumper, J. *et al.* Highly accurate protein structure prediction with AlphaFold. *Nature* **596**, 583–589 (2021).
10. Varadi, M. *et al.* AlphaFold Protein Structure Database: massively expanding the structural coverage of protein-sequence space with high-accuracy models. *Nucleic Acids Res.* **50**, D439–D444 (2022).
11. Benjamini, Y., Krieger, A. M. & Yekutieli, D. Adaptive linear step-up procedures that control the false discovery rate. *Biometrika* **93**, 491–507 (2006).
12. Choi, J. Y., Jang, T.-H. & Park, H. H. The mechanism of folding robustness revealed by the crystal structure of extra-superfolder GFP. *FEBS Lett.* **591**, 442–447 (2017).
13. Stepanenko, O. V. *et al.* Distinct Effects of Guanidine Thiocyanate on the Structure of Superfolder GFP. *PLoS ONE* **7**, e48809 (2012).
14. Pédelacq, J.-D., Cabantous, S., Tran, T., Terwilliger, T. C. & Waldo, G. S. Engineering and characterization of a superfolder green fluorescent protein. *Nat. Biotechnol.* **24**, 79–88 (2006).
15. Yeong, V., Wang, J., Horn, J. M. & Obermeyer, A. C. Intracellular phase separation of globular proteins facilitated by short cationic peptides. *Nat. Commun.* **13**, 7882 (2022).
16. Yoshida, T. *et al.* Compartmentalization of soluble endocytic proteins in synaptic vesicle clusters by phase separation. *iScience* **26**, 106826 (2023).
17. Ogunmowo, T. *et al.* *Intersectin and Endophilin condensates prime synaptic vesicles for release site replenishment.* <http://biorxiv.org/lookup/doi/10.1101/2023.08.22.554276> (2023) doi:10.1101/2023.08.22.554276.
18. Hoffmann, C. *et al.* Synapsin condensation controls synaptic vesicle sequestering and dynamics. *Nat. Commun.* **14**, 6730 (2023).
19. Park, D. *et al.* Cooperative function of synaptophysin and synapsin in the generation of synaptic vesicle-like clusters in non-neuronal cells. *Nat. Commun.* **12**, 263 (2021).
20. Courtney, K. C. *et al.* Synaptotagmin 1 oligomerization via the juxtamembrane linker regulates spontaneous and evoked neurotransmitter release. *Proc. Natl. Acad. Sci.* **118**, e2113859118 (2021).
21. Ulianov, S. V. *et al.* Suppression of liquid–liquid phase separation by 1,6-hexanediol partially compromises the 3D genome organization in living cells. *Nucleic Acids Res.* **49**, 10524–10541 (2021).

REVIEWERS' COMMENTS

Reviewer #1 (Remarks to the Author):

I find the revisions have significantly strengthened the overall quality of the paper. I have no further comments to add.

Reviewer #2 (Remarks to the Author):

The authors have addressed all of my previous comments. Especially, the Ca²⁺ experiments that the authors newly performed are very interesting. I now recommend publication.

Congratulations!